# An adaptive synaptic array using Fowler–Nordheim dynamic analog memory

Darshit Mehta [1,3], Mustafizur Rahman [2,3], Kenji Aono [2] & Shantanu Chakrabartty [1,2✉]

In this paper we present an adaptive synaptic array that can be used to improve the energy-efficiency of training machine learning (ML) systems. The synaptic array comprises of an ensemble of analog memory elements, each of which is a micro-scale dynamical system in its own right, storing information in its temporal state trajectory. The state trajectories are then modulated by a system level learning algorithm such that the ensemble trajectory is guided towards the optimal solution. We show that the extrinsic energy required for state trajectory modulation can be matched to the dynamics of neural network learning which leads to a significant reduction in energy-dissipated for memory updates during ML training. Thus, the proposed synapse array could have significant implications in addressing the energy-efficiency imbalance between the training and the inference phases observed in artificial intelligence (AI) systems.

[1] Department of Biomedical Engineering, Washington University in St. Louis, St. Louis, MO, USA. [2] Department of Electrical and Systems Engineering, Washington University in St. Louis, St. Louis, MO, USA. [3]These authors contributed equally: Darshit Mehta, Mustafizur Rahman.
✉email: shantanu@wustl.edu

I mplementation of reliable and scalable synaptic weights or memory remains an unresolved challenge in the design of energy-efficient ML and neuromorphic processors[1]. Ideally, the synaptic weights should be "analog" and should be implemented on a non-volatile, and yet easily modifiable storage device[2]. Furthermore, if these memory elements are integrated in proximity with the computing circuits or processing elements, then the resulting compute-in-memory (CIM) architecture[3,4] has the potential to mitigate the "memory wall"[5–7] which refers to the energy-efficiency bottleneck in ML processors that arises due to repeated memory access. In most practical and scalable implementations, the processing elements are implemented using CMOS circuits; as a result, it is desirable that the analog synaptic weights be implemented using a CMOS-compatible technology. In literature, several multi-level non-volatile memory devices have been proposed for implementing analog synapses. These include two-terminal memristive devices such as resistive random-access memories (RRAM)[8], magnetic random-access memories (MRAM)[9], Phase Change Memory (PCM)[10], Spin-Transfer Torque Magnetic RAM (STT-MRAM)[11], Conductive Bridge RAM[12] or the three terminal devices like the floating-gate transistors[13], ferroelectric field-effect transistor-based memory (FeFET)[14], Charge Trap Memory[15] and Electrochemical RAMs (ECRAM)[16]. In all of these devices, the analog memory states are static in nature, where each of the states needs to be separated from others by an energy barrier $\Delta E$. For example, in RRAM devices the state of the conductive filament between two electrodes determines the stored analog value, whereas in charge-based devices like floating-gates or FeFET, the state of polarization determines the analog value. To ensure non-volatile storage, it is critical that the energy-barrier $\Delta E$ is chosen to be large enough to prevent memory leakage due to thermal fluctuations and other environmental disturbances. However, the height of the energy barrier $\Delta E$ also sets the fundamental limit on the energy dissipated to switch between different analog storage states. For example, switching the RRAM memory state requires 100 fJ per bit[17], whereas STT-MRAM requires about 4.5pJ per bit[18]. A learning/training algorithm that adapts the stored weights in quantized steps $(..., W_{n-1}, W_n, W_{n+1}, ...)$ so as to minimize a system-level loss-function $L(W)$, as shown in Fig. 1(a), has to dissipate a minimum energy of $(..., \Delta E_{n-1}, \Delta E_n, \Delta E_{n+1}, ...)$ for memory updates. Separating the static states by an energy-barrier also allows the learning algorithm to precisely control the parameter retention time (parameter leakage) between subsequent parameter updates, however, this mode of updates do not exploit the physics of learning to optimize for energy-efficiency. In many energy-efficient ML training formulations, and in particular analog ML systems, the loss-function $L(W)$ is represented by an equivalent energy-functional of a physical ML system[19] and learning/training involves a natural evolution of the system dynamics towards the minimum energy (optimal) state based on input stimuli (or equivalently training data). Thus, the physics of the system evolution process selects the minimum energy path towards the desired optimum. A synaptic element that is matched to this system dynamics needs to be adaptive with respect to its memory retention time which can then be traded-off with respect to the energy-dissipation per update.

In this paper, we present such a synaptic element that uses dynamical states (instead of static states) to implement analog memory and is matched to the dynamics of ML training. The core of the proposed device is itself a micro-dynamical system and the system-level learning/training process modulates the dynamical state (or state trajectory) of the memory ensembles. The concept is illustrated in Fig. 1b, which shows a reference ensemble trajectory that continuously decays towards a reference *zero vector* without the presence of any external modulation. However,

during the process of learning, the trajectory of the memory ensemble is pushed towards an optimal solution $W^*$. The main premise of this paper is that the extrinsic energy $(..., \Delta E_{n-1}, \Delta E_n, \Delta E_{n+1}, ...)$ required for modulation, if matched to the dynamics of learning, could reduce the energy-budget for ML training. This is illustrated in Fig. 1c which shows a convergence plot corresponding to a typical ML system as it transitions from a training phase to an inference phase. During the training phase, the synaptic weights are adapted based on some learning criterion whereas in the inference phase the synaptic weights remain fixed or are adapted intermittently to account for changes in the operating conditions. As a result, during the training phase the number of weight updates is significantly higher than during the inference phase. Take for example support-vector machine (SVM) training, the number of weight updates scale quadratically with the number of support vectors and the size of the training data, whereas adapting the SVM during inference only scales linearly with the number of support-vectors[20]. Thus, for a constant energy dissipation per update, the total energy-dissipated due to weight updates is significantly higher in training than during inference. However, if the energy-budget per weight updates could follow a temporal profile as shown in Fig.1c, wherein the energy dissipation is no longer constant, but inversely proportional to the expected weight-update rate, then the total energy dissipated during training could be significantly reduced. One way to reduce the synaptic weight update or memory write energy budget is to trade-off the weight's retention rate according to the profile shown in Fig. 1c. The desired retention rate profile could then be achieved by adaptively changing the energy-barrier height as shown in Fig. 1c - inset. During the training phase, the synaptic element can tolerate lower retention rates or parameter leakage because this physical process could be matched to the process of weight decay or regularization, a technique commonly used in ML algorithms to achieve better generalization performance[21]. As shown in Fig. 1c, the synapse's retention rate should increase as the training progresses such that at convergence or in the inference phase the weights are stored as a non-volatile memory.

In this paper, we describe a dynamic analog memory (DAM) that can exhibit a temporal profile similar to that of Fig. 1c. Furthermore, the memory is implemented on a standard CMOS process without the need for any additional processing layers. Figure 1d shows a micrograph of a DAM array where each element of the array implements the circuit shown in Fig. 1e. In the Supplementary section I, we provide additional details for implementing the circuit of Fig. 1e in a standard CMOS process. The proposed DAM requires a Fowler–Nordheim (FN) quantum-tunneling barrier which can be created by injecting sufficient electrons onto a polysilicon island (floating-gate) that is electrically isolated by thin silicon-di-oxide barriers[22]. As the electron tunnels through the triangular barrier, as shown in Fig. 1f, the barrier profile changes which further inhibits the tunneling of electrons. We have previously shown that the dynamics of this simple system is robust enough to implement time-keeping devices[23] and self-powered sensors[24]. In this paper, we use a pair of synchronized FN-dynamical systems to implement a DAM suitable for implementing ML training/inference engines. Figure 1g shows the dynamics of two FN-dynamical systems, labeled as SET and RESET, whose analog states continuously and synchronously decay with respect to time. In our previous work[22,24], we have shown the dynamics across different FN-dynamical systems can be synchronized with respect to each other with an accuracy greater than 99.9%. However, when an external voltage pulse modulates the SET system, as shown in Fig. 1g, the dynamics of the SET system becomes desynchronized with respect to the RESET system. The degree of desynchronization is

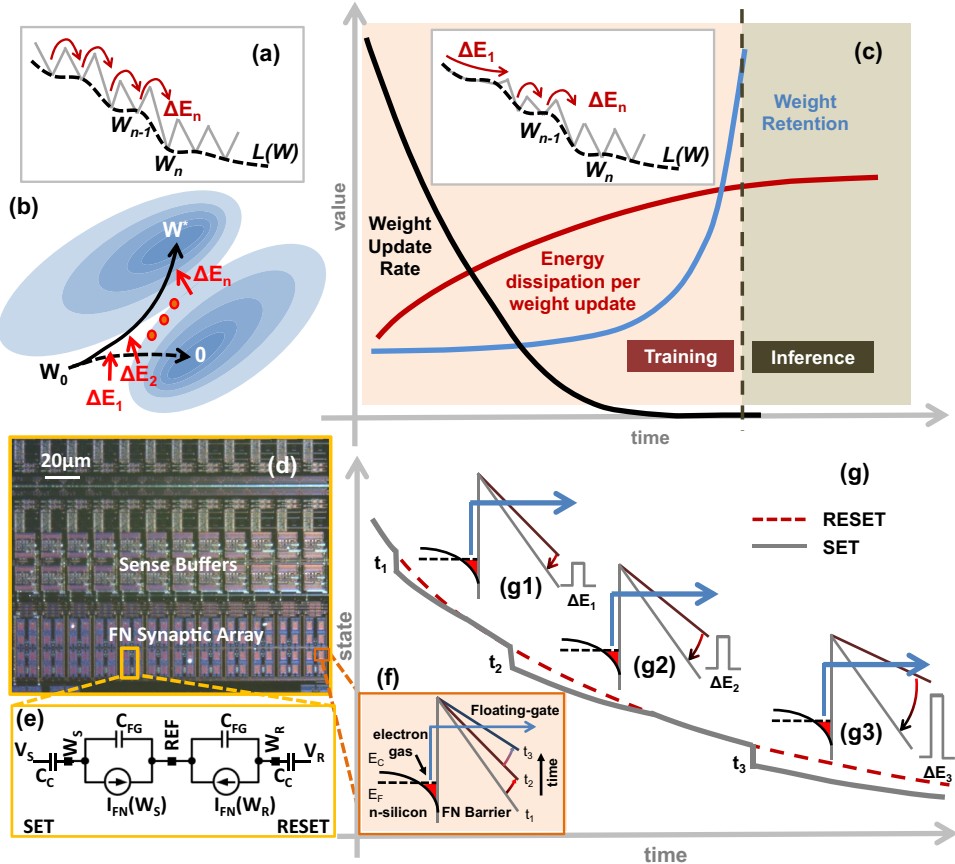

**Fig. 1 Motivation and principle of operation for the proposed synaptic memory device. a** Conventional non-volatile analog memory where transition between analog static states ($W_{n-1}$, $W_n$) dissipates energy ($\Delta E_n$); **b** dynamic analog memory where an external energy ($\Delta E_1$, $\Delta E_2 \dots \Delta E_n$) is used to modulate the trajectory of the memory states ($W_O$) towards the optimal solution ($W^*$); **c** desired analog synapse characteristic where the memory retention rate is traded-off with the write energy; reducing the energy dissipation per weight update in training phase by matching the dynamics of the dynamic analog memory to the weight decay as shown in (inset) where the height of the energy barrier ($\Delta E_n$) increases as the training progresses (**d**) micrograph of a fabricated DAM array along with (**e**) its equivalent circuit where the leakage current $I_{FN}$ is implemented by (**f**) the electron transport across a Fowler–Nordheim (FN) tunneling barrier; **g** implementation of the FN tunneling based DAM where dynamic states g1–g3 determines the energy dissipated ($\Delta E_1$, $\Delta E_2$, $\Delta E_3$) per memory update at time instance $t_1$–$t_3$ and memory retention rate.

a function of the state of the memory at different time instances (Fig. 1g, insets g1–g3) which determines the memory's retention rate. For instance, at time-instant $t_1$, a small magnitude pulse would produce the same degree of desynchronization as a large magnitude pulse at the time-instant $t_3$. However, at $t_1$ the pair of desynchronized systems (SET and RESET) would resynchronize more rapidly as compared to desynchronized systems at time-instants $t_2$ or $t_3$. This resynchronization effect results in shorter data retention; however, this feature could be leveraged to implement weight-decay in ML training. At time-instant $t_3$, the resynchronization effect is weak enough that the FN-dynamical system acts as a persistent non-volatile memory with high data-retention time. In the Methods section, we describe how the FN-dynamical system mathematical model can be matched to ML training formulation and the weight-decay dynamics required for learning and generalization. The model also shows that the voltage or energy required for updating the memory can be annealed according to the profile shown in Fig. 1c.

## Results

**Dynamic analog memory with an asymptotic non-volatile storage.** The dynamics of the FN-tunneling-based DAM (or FN-DAM) were verified using prototypes fabricated in a standard CMOS process (micrograph shown in Fig. 1d). The FN-DAM devices were programmed and initialized through a combination

of FN tunneling and hot electron injection. A detailed description of the general programming process can be found in[24] and in the "Methods" section we describe implementation specific to this work. The tunneling nodes ($W_S$ and $W_R$ in Fig. 1e) were initialized to around 8 V and decoupled from the readout node by a decoupling capacitor to the sense buffers (shown in supplementary information Fig. 1). The readout nodes were biased at a lower voltage (~3 V) to prevent hot electron injection[25] onto the floating-gate during the readout operation. The capacitive decoupling of the read-out circuitry from the memory also reduces the effect of read disturbances and in Supplementary Fig. 2, we show measurement results that verify that the effect of read disturbance is random and the magnitude of the disturbance is less than the precision of the memory update and read-out circuits.

Figure 2 shows the measured dynamics of the FN-DAM device in different initialization regimes used in ML training, as described in Fig. 1g. The different regimes were obtained by initializing the tunneling nodes ($W_S$ and $W_R$) to different voltages (see "Methods" section), whilst ensuring that the tunneling rates on the $W_S$ and $W_R$ nodes were equal. Initially (during the training phase), tunneling-node voltages were biased high (readout node voltage of 3.1 V), leading to faster FN tunneling (Fig. 2, inset a1). A square input pulse of 100 mV magnitude and 500 ms duration (5 fJ of input energy) was found to be sufficient to desynchronize

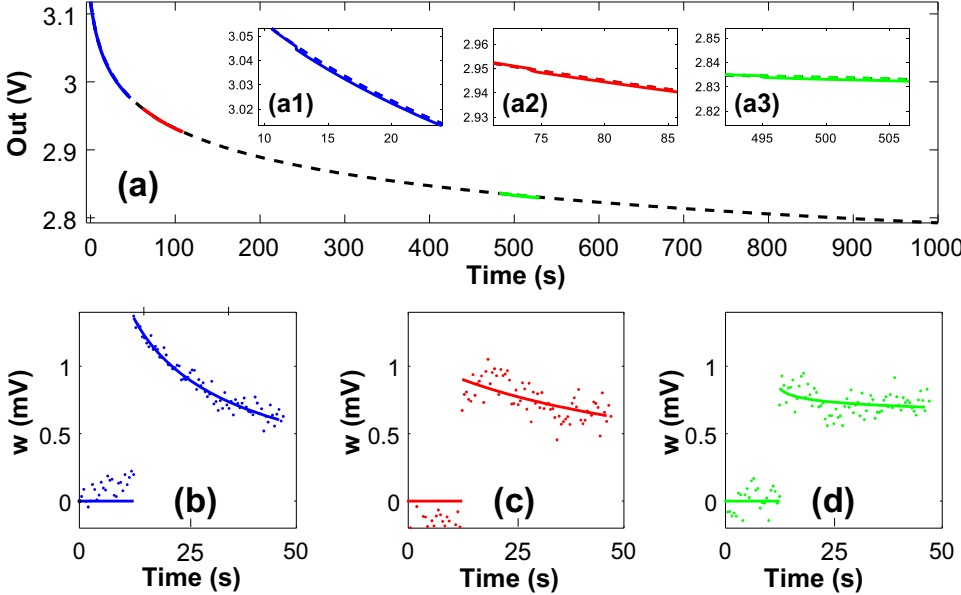

**Fig. 2 FN-DAM plasticity under different operating regions. a** $W_S$ (solid line) and $W_R$ (dashed line) response under three different operating regimes (zoomed insets: a1, a2, a3) determined by FN-DAM initialization voltage. **b–d** FN-DAM response ($w$) calculated as difference between $W_S$ and $W_R$ voltage values in the three regimes demonstrating different plasticity. Dots are measured datapoints while lines correspond to fit to the data.

the SET node by 1 mV. This desynchronization, $w = (W_S - W_R)$, stores the state of the dynamical analog memory. However, as shown in Fig. 2b, the rate of resynchronization in this regime is high, which leads to a decay in the stored weight down to 30% in 40 s. At $t = 90$ s, the voltage at node $W_S$ has reduced (readout node voltage of 2.9 V shown in Fig. 2, inset a2), and a larger voltage amplitude (500 mV) is required to achieve the same desynchronization magnitude of 1 mV. This corresponds to an energy expenditure of 125 fJ. However, as shown in Fig. 2c, the rate of resynchronization is low in this regime, leading to a decay in the stored weight down to 70% its value in 40 s. Similarly, at a later time instant $t = 540$ s (Fig. 2, inset a3), a 1 V signal desynchronizes the recorder by 1 mV, and as shown in Fig. 2d, in this regime 95% of the stored weight value is retained after 40 s. This mode of operation is suitable during the inference phase of machine learning when the weights have already been trained, but the models need to be sporadically adapted to account for statistical drifts. Modeling studies described in Supplementary section III show that the write energy per update starts from as low as 5 fJ and increases to 2.5 pJ over a period of a period of 12 days. During the same time, the memory becomes less plastic with the increase in the memory retention time as shown in Supplementary section IV. Asymptotically, the FN-DAM exhibits retention times similar to that of other FLASH-based memory.

The next set of experiments verified if the analog state of an FN-DAM device can be adapted (incremented or decremented) using digital pulses (using a digital logic or a spiking neuron). Each of the differential DAM elements in the FN-DAM device was programmed by independently modulating the SET and RESET junctions shown in Fig. 1e. The corresponding $W_S$ and $W_R$ nodes were initially synchronized with respect to each other. After a programming pulse was applied to the SET or RESET control gate, the difference between the voltages at the $W_S$ and $W_R$ nodes was measured using an array of sense buffers. In results shown in Fig. 3a–d, a sequence of 100 ms-long 3 V SET and RESET pulses was applied. The measured difference between the voltages at the $W_S$ and $W_R$ nodes indicates the current state of the memory. Each SET pulse increments the state while a RESET pulse decrements the state. In this way, the FN-device can implement a DAM that is bidirectionally programmable with

unipolar pulses. Note that, unlike conventional FLASH memory, the magnitude of the programming pulse is significantly lower. Figure 3d also shows the cumulative nature of the FN-DAM updates which implies that the device can work as an incremental/decremental counter.

Figure 3e, f show measurement results that demonstrate the resolution at which an FN-DAM can be programmed as an analog memory. The analog state can be updated by applying digital pulses of varying frequency and variable number of pulses. In Fig. 3e, four cases of applying a 3 V SET signal for a total of 100 ms are shown: a single 100 ms pulse; two 50 ms pulses; four 25 ms pulses; and eight 12.5 ms pulses. The results show the net change in the stored weight was consistent across the 4 cases. A higher frequency leads to a finer control of the analog memory updates. Note that any variations across the devices can be calibrated or mitigated by using an appropriate learning algorithm[26]. The variations could also be reduced by using careful layout techniques and precise timing of the control signals.

**Characterization of FN-DAM.** The FN-DAM device can be programmed by changing the magnitude of the SET/RESET pulse or its duration (equivalently number of pulses of fixed duration). Figure 4a shows response when the magnitude of the SET and RESET input signals varies from 4.1 to 4.5 V. The measured response shown in Fig. 4a shows an exponential relationship with the amplitude of the signal. When short-duration (10 ms) pulses are used for programming, the stored value varies linearly with the number of pulses, as shown in Fig. 4b. However, repeated application of pulses with constant magnitude produces a successively smaller change in programmed value due to the dynamics of the DAM device (Fig. 4c). One way to achieve a constant response is to pre-compensate the SET/RESET control voltages such that a target voltage difference $w = (W_S - W_R)$ can be realized. The differential architecture increases the device state robustness against disruptions from thermal fluctuations (Fig. 4d). The stored value on DAM devices will leak due to thermal-induced processes or due to trap-assisted tunneling. However, in DAM, the weight is stored as the difference in the

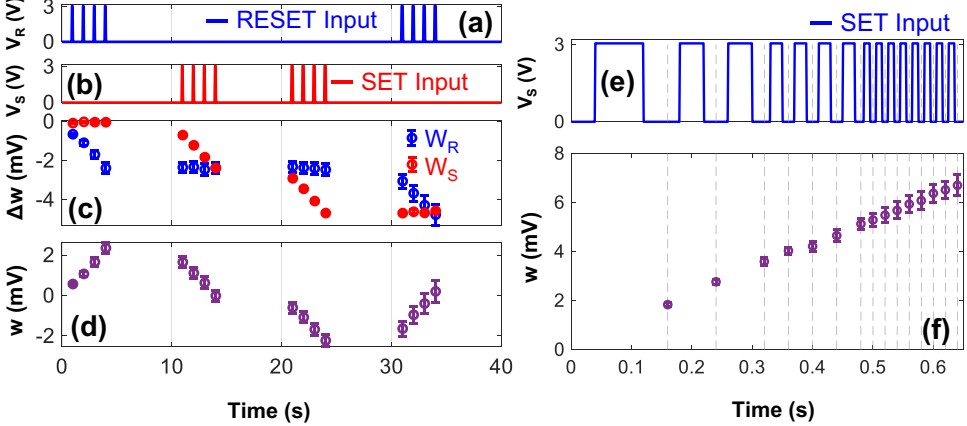

**Fig. 3 Memory update characterization. a**, **b** SET and RESET input sequence. **c** Change in $W_S$ and $W_R$ potentials due to SET and RESET pulses. **d** DAM response calculated as difference between $W_S$ and $W_R$ voltages. **e**, **f** FN-DAM response to SET pulses of varying frequency. Error bars indicate standard deviation estimated across 12 devices.

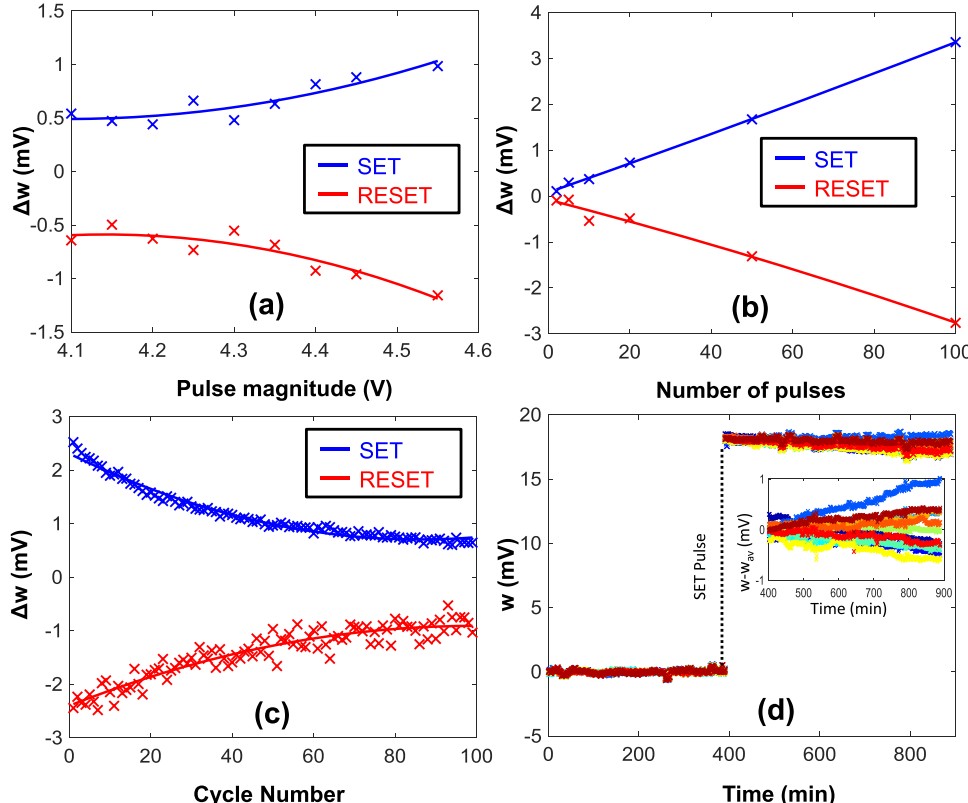

**Fig. 4 Device characterization. a** DAM response to pulses of different magnitude but same duration (10 ms). **b** DAM response to varying number of pulses of 4 V amplitude and 10 ms duration. **c** Change in DAM response with each pulse of same magnitude (4 V) and duration (10 ms). **d** FN-DAM response measured at 100 °C when a SET pulse is applied to 12 different FN-DAM elements (each color corresponds to a different memory element).

voltages corresponding to $W_S$ and $W_R$ tunneling junctions which are similarly affected by temperature fluctuations. This is shown in Fig. 4d where the FN-DAM array was programmed/operated at 100 °C and the dynamic response was measured over a duration of 15 h. The baseline drift due to the memory read-out circuits were first calibrated during the first 400 min and used for zeroing out the dynamical response of each of the FN-DAM device. Then, at 400 min time instant a SET pulse (3.3 V for 1 s duration) was applied to all the FN-DAM devices which programmed all the devices to a specific memory state. The degree of desynchronization was continuously measured and is plotted in Fig. 4d. Over a duration of 8 h the drift in the stored analog value

is less than 10%. This result could also be used to estimate the memory retention time as described in the Supplementary Information section IV, which is expected to vary depending on the current state of the memory.

**FN-DAM based co-design of classifiers and neural networks.** We first experimentally demonstrate the benefits of FN-DAM-based weights when training a simple linear classifier. For these results, two FN-DAM devices were independently programmed according to the perceptron training rule[27]. We trained the weights of a perceptron model to classify a linearly separable

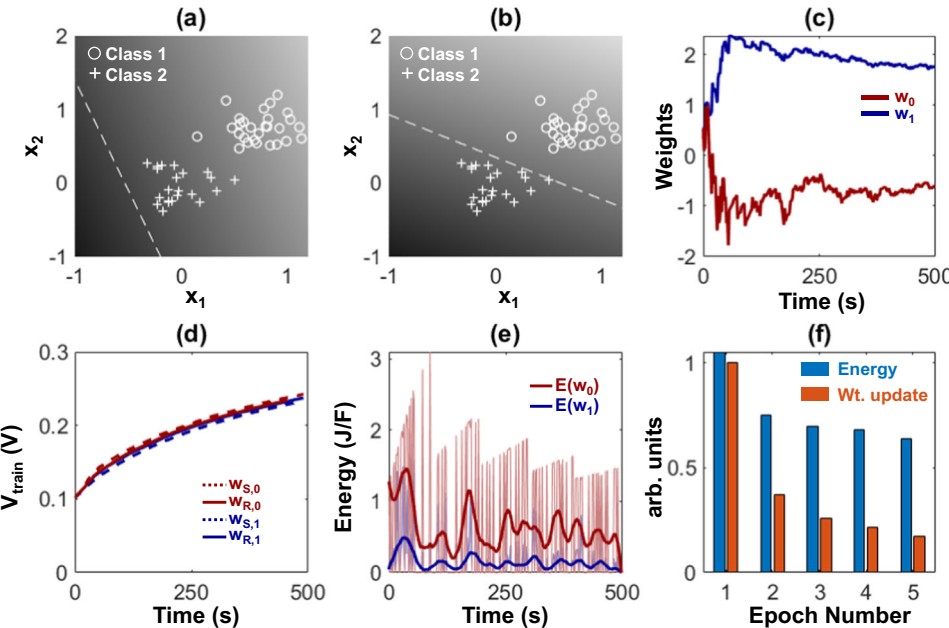

**Fig. 5 Synaptic memory for neuromorphic applications. a** Test data set with randomly initialized decision boundary. **b** Decision boundary after training. **c** Evolution of weights ($w_0$ and $w_1$) after 5 epochs. **d** Input voltage required for initiating a unit change in weights ($W_{s,0}$, $W_{R,0}$, $W_{s,1}$, $W_{R,1}$). **e** Energy ($E(w_0)$, $E(w_1)$) expended in updating the weights ($w_0$ and $w_1$). **f** Average magnitude of weight update and average energy required for each epoch.

dataset comprising of 50 instances of two-dimensional vectors, as shown in Fig. 5a. During each epoch, the network loss function and gradients were evaluated for every training point in a randomized order, with time interval between successive training points being two seconds. Figure 5b shows that after training for 5 epochs, the learned boundary can correctly classify the given data. Figure 5c shows the evolution of weights as a function of time. As can be noted in the figure, initially the magnitude of weight updates (negative of the cost function gradient) was high for the first 50 s, after which the weights stabilized and required smaller updates. The energy consumption of the training algorithm can be estimated based on the magnitude and number of the SET/ RESET pulses required to carry out the required update for each misclassified point. As the SET/RESET nodes evolve in time, they require larger voltages for carrying out updates, shown in Fig. 5d. The gradient magnitude was mapped onto an equivalent number of 1 kHz pulses, rounding to the nearest integer. Figure 5e shows the energy (per unit capacitance) required to carry out the weight update whenever a point was misclassified. Though the total magnitude of weight update decreased with each epoch, the energy required to carry out the updates had lower variation (Fig. 5f). The relatively larger energy required for smaller weight updates at later epochs led to longer retention times of the weights. Similar energy-dissipation and weight update profiles were also obtained when a larger FN-DAM array is used to store the training parameters of a three-layer neural network implementing a multi-layer perceptron (MLP). Details of the network architecture and training procedure are described in the Methods section and in Supplementary Information section VII. Figure 6a–c shows the FN-DAM training dynamics when the MLP neural network is trained on the Fisher Iris dataset[28]. In particular, Fig. 6c shows that by adapting the programming pulses, the energy-dissipation profile across training and inference can be equalized, as was proposed in Fig. 1c. The dynamical systems model summarized in the Methods section can also be used to evaluate the energy-efficiency gains that can be obtained by co-designing a convolutional neural network (CNN) training engine using FN-DAMs. Details of the CNN architecture are

provided in the supplementary information section VII. The result of the co-design is shown in Fig. 6d and in Table 1 where we show that an FN-DAM-based deep neural network (DNN) can achieve similar classification accuracy as a conventional DNN. To compare the energy dissipation of the FN-DAM neural network implementation, we used a RRAM energy per bit dissipation metric (100 fJ/bit)[17] and for an FN-DAM implementation we used an energy dissipation model described in Supplementary Information section III. Note that amongst CMOS-compatible non-volatile analog memories RRAM is one of the most energy-efficient synapses. The result in Fig. 6e shows that training an FN-DAM-based neural network dissipates significantly lower energy compared to the RRAM-based neural network. Note that for this demonstration, only the fully connected layers were trained while the feature layers were kept static. This mode of training is common for many practical DNN implementations on edge computing platforms where the goal is not only to improve the energy-efficiency of inference but also for training[29]. The results in Fig. 6e and Table 1 show that the neural network training and accuracy are robust even when mismatch is introduced into the FN-DAM model. Details of the mismatch model are provided in the "Methods" section.

## Discussions

In this paper, we reported a FN quantum tunneling-based DAM (FN-DAM) whose physical dynamics can be matched to the dynamics of weight updates used in ML or neural network training. During the training phase, the weights stored on FN-DAM are plastic in nature and decay according to a learning-rate evolution that is necessary for the convergence of the gradient-descent training[30]. As the training phase transitions to an inference phase, the FN-DAM acts as a non-volatile memory. As a result, the trained weights are persistently stored without requiring any additional refresh steps (used in volatile embedded DRAM architectures[31]). The plasticity of FN-DAM during the training phase can be traded off with the energy required to update the weights. This is important because the number of

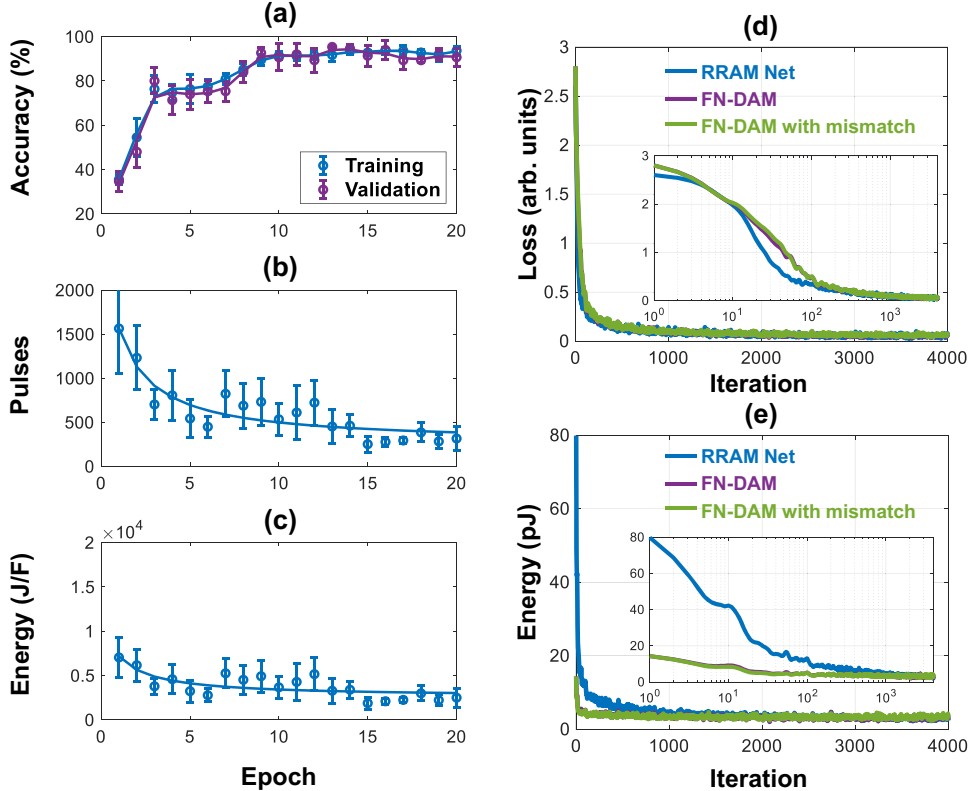

**Fig. 6 Synaptic memory for deep neural network tasks. a–c** Experimental training on Fisher Iris dataset over five trials: **a** Five-fold cross-validation accuracy of model over 20 epochs for training set (120 points) and validation set (30 points). **b** Total pulses required in implementing weight update for entire synaptic array during each epoch. **c** Energy per unit capacitance expended in updating the weights. Note: scale of *Y*-axis is set to match that of panel (**b**). Error bars in **a–c** indicate standard deviation estimated across five trials. **d**, **e** Simulated training on MNIST dataset: **d** Network loss for three types of network models. Inset shows same data with *x*-axis in log scale. **e** Energy dissipated in updating the network weights for three types of network models. Inset shows same data with *X*-axis in log scale.

| Table 1 Test accuracy obtained for different variants of CNNs on the MNIST task. | | |
|---|---|---|
| **Accuracy (%)** | **After 9 epochs** | **After 10 epochs** |
| Standard CNN | 98.9 | 98.9 |
| FN-DAM CNN | 98.9 | 99.2 |
| FN-DAM CNN with mismatch | 97.9 | 97.3 |

weight updates during training scale quadratically with the number of parameters, hence the energy budget during training is significantly higher than the energy budget for inference. The dynamics of FN-DAM bears similarity to the process of annealing used in neural network training and other stochastic optimization engines to overcome local minima artifacts[32]. Thus, it is possible that FN-DAM implementations or ML processors can naturally implement annealing without dissipating any additional energy. If such dynamics were to be emulated on other analog memories, it would require additional hardware and control circuitry.

Several challenges exist in scaling the FN-DAM to large neural-networks. Training a large-scale neural network could take days to months[33] depending on the complexity of the problem, complexity of the network, and the size of the training data. This implies that the FN-DAM dynamics need to match the long training durations as well. Fortunately, the *1/log* characteristics of FN devices ensure that the dynamics could last for durations greater than a year[34]. The other challenge that might limit the scaling of FN-DAM to large neural network is the measurement

precision. The resolution of the measurement and the read-out circuits limit the energy-dissipated during memory access and how fast the gradients can be computed (Supplementary Information Fig. 5). For instance, a 1 pF floating-gate capacitance can be initialized to store $10^7$ electrons. Even if one were able to measure the change in synaptic weights for every electron tunneling event, the read-out circuits would need to discriminate 100 nV changes. A more realistic scenario would be measuring the change in voltage after 1000 electron tunneling events which would imply measuring 100 μV changes. However, this will reduce the resolution of the stored weights/updates to 14 bits. This resolution might be sufficient for training a medium-sized neural network; however, it is still an open question if this resolution would be sufficient for training large-scale networks[35,36]. A mechanism to improve the dynamic range and the measurement resolution is to use a current-mode readout integrated with current-mode neural network architecture. If the read-out transistor is biased in weak-inversion, 120 dB of dynamic range could be potentially achieved. However, note that even in this operating mode, the resolution of the weight would still be limited by the number of electrons and the quantization due to electron transport. Addressing this limitation would be a part of future research.

If the proposed FN-DAM were to be used as a static analog memory, then measuring 1 mV differences to distinguish between different memory states would be challenging, especially if device mismatch were to be taken into account. However, the analog value stored on the FN-DAM array is updated within a learning loop that minimizes a system level objective function (cumulative

loss or distance). Thus, the effect of any static mismatch across the memory cells gets calibrated out during the process of training. The important aspect for the calibration process to be successful is that the memory update be monotonic with respect to error-gradient and the precision of the updates be high enough (typically greater than 12 bits). Both of these requirements are met by FN-DAM due to the physics of electron tunneling. In fact, the effect of calibration due to learning can be seen in the FN-DAM neural network training (Fig. 6d, e) where the classification accuracy is independent of the initial choice of the FN-DAM state and the mismatch in FN-DAM device characteristics. The effect of blurring, due to the presence of thermal noise is in fact beneficial for training the neural network since it helps in overcoming artifacts due to local minima. Once the FN-DAM has transitioned to a non-volatile state (during inference), the effect of blurring is significantly reduced as the energy barrier separating different analog states is significantly higher than energy due to thermal fluctuations. However, the effect of blurring due to measurement noise needs to be compensated by averaging or increasing the cumulative measurement time.

In this work, we have used a voltage buffer (source follower) to the read the state of the FN-DAM cell. However, a current mode readout could also be used to differentiate mV changes in FN-DAM voltages. In particular, if the read-out transistor is biased in weak-inversion, then the exponential dependence between the gate voltage and the drain current could be used amplify the change in voltage. We have previously used this method in[37] for floating-gate current memory arrays and in[38] we reported an active feedback-based approach to improve the resolution of the voltage-mode read-out. However, in both these implementations there will exist a trade-off with the accuracy of the measurement resolution and the read-out speed[39].

Another limitation that arises due to finite number of electrons stored on the floating-gate and transported across the tunneling barrier during SET and RESET, is the speed of programming. Shorter duration programming pulses would reduce the change in stored voltage (weight) which could be beneficial if precision in updates is desired. In contrast, by increasing the magnitude of the programming pulses, as shown in Fig. 4a, the change in stored voltage can be coarsely adjusted. However, this would limit the number of updates before the weights saturate. Note that due to device mismatch the programmed values would be different on different FN-DAM devices.

In terms of endurance, after a single initialization, the FN-DAM can support $10^3$–$10^4$ update cycles before the weight saturates. However, at the core FN-DAM is a FLASH technology and could potentially be reinitialized again. Given that the endurance of FLASH memory is $10^3$ [8], it is anticipated that FN-DAM to have an endurance of $10^6$–$10^7$ cycles. In terms of other memory performance metrics, the $I_{ON}/I_{OFF}$ ratio for the FN-DAM is determined by the operating regime and the read-out mechanism. Supplementary Fig. 5 shows the expected ratio estimated using the FN-DAM model. Also, FN-DAM when biased as a non-volatile memory requires on-chip charge-pumps only to generate high-voltage programming pulses for infrequent global erase; thus, compared to FLASH memory, FN-DAM should have fewer failure modes[9]. Since FN-DAM can also be implemented on conventional FLASH memories, the synapses could be scaled to future 3-D and 2.5D FLASH processes where high synaptic densities can be achieved for implementation of large-scale neural networks.

The main advantage of FN-DAM compared to other emerging memory technologies is its scalability and compatibility with CMOS. At its core, FN-DAM is based on floating-gate memories which have been extensively studied in the context of machine learning architectures[13]. Furthermore, from an equivalent circuit

point of view, FN-DAM could be viewed as a capacitor whose charge can be precisely programmed using CMOS processing elements. Due to its unique decay characteristics, FN-DAM also provides a balance between weight-updates that are not too small so that learning never occurs versus weight-updates being too large such that the learning becomes unstable. The physics of FN-DAM ensures that weight decay (in the absence of any updates) towards a zero vector (due to resynchronization) which is important for neural network generalization[40]. For implementing a large-scale neural network, the FN-DAM form-factor would be required to be reduced which would affect device variability and mismatch[41–44]. However, in our prior work,[23,34] we have shown that the dynamics of the FN-DAM cell (in steady-state) is determined primarily by the gate-oxide thickness, a parameter that is very well controlled across processes. An oxide thickness greater than 10 nm ensures that the electron-leakage mechanism is dominated by FN quantum tunneling (instead of direct quantum tunneling). Thus, FN-DAM devices should be implementable on most sub-10nm CMOS processes that allow fabrication of thicker gate-oxide transistors for input/output devices.

Like other analog non-volatile memories, FN-DAM could be used in any previously proposed CIM architectures. However, in conventional CIM implementations the weights are trained off-line and then downloaded on chip without retraining the processor[1]. This makes the architecture prone to analog artifacts like offsets, mismatch, and non-linearities. On-chip learning and training mitigates this problem whereby the weights self-calibrate for the artifacts to produce the desired output[2]. However, to support on-chip training/learning, weights need to be updated at a precision greater than 12 bits[36]. In this regard, FN-DAM exhibits a significant advantage compared to other analog memories. Even though in this proof-of-concept work, we have a used a hybrid chip-in-the-loop training paradigm, it is anticipated that in the future the training circuits and FN-DAM modules could be integrated together on-chip.

From a neuromorphic point of view, FN-DAMs could be used to mimic network level synaptic adaptation or pruning which plays a pivotal role in determining the optimal network configuration during the process of learning. For instance, it has been reported that a child's brain has significantly denser connectivity than an adult brain[3] and consumes 50% of the body's resting energy metabolism (BMR). Years of learning and synaptic pruning produces a network that tends towards optimality in terms of both energy and performance in adulthood, when the brain accounts for only 20% of the BMR[3]. The adaptability of the proposed FN-DAM could be used to mimic this effect in artificial machine learning systems.

If the FN-DAM updates are driven by constant voltage pulses (or fixed energy pulses like spikes) then the memory could be used to emulate the ageing effects in synaptic plasticity that is observed in neurobiological systems[4]. Like biological synapses, the relative change in value stored on FN-DAM or synaptic efficacy reduces with time for the same magnitude of applied input voltage pulses (or stimuli)[45]. Exploiting this feature of the FN-DAM to mimic neurobiologically relevant synaptic dynamics in artificial neural networks would also be a topic of future research.

## Methods

**Initialization of the FN-DAM array.** For each node of each recorder, the readout voltage was programmed to around 3 V while the tunneling node was operating in the tunneling regime (Supplementary Fig. 1). This was achieved through a combination of tunneling and injection. Specifically, $V_{DD}$ was set to 7 V, input to 5 V, and the program tunneling pin was gradually increased to 23 V. Around 12–13 V the tunneling node's potential would start increasing. The coupled readout node's potential would also increase. When the readout potential went over 4.5 V, electrons would start injecting into the readout floating gate, thus ensuring its potential

was clamped below 5 V. After this initial programming, VDD was set to 6 V for the rest of the experiments. See Supplementary section I for further details. After one-time programming, input was set to 0 V, input tunneling voltage was set to 21.5 V for 1 min and then the floating gate was allowed to discharge naturally. Readout voltages for the SET and RESET nodes were measured every 500 ms. The rate of discharge for each node was calculated, and a state where the tunneling rates would be equal was chosen as the initial synchronization point for the remainder of the experiments.

**FN tunneling dynamics**. The FN tunneling current is a function of the floating-gate capacitance $C_T$ and the floating-gate voltage $V(t)$ and is given by:

$$I_{FN}(V(t)) = C_T \frac{d(V(t))}{dt} = C_T \left(\frac{k_1}{k_2}\right) V^2 \exp\left(-\frac{k_2}{V}\right) \quad (1)$$

where $k_1$ and $k_2$ are device specific parameters. Solving (1) leads to the floating-gate voltage $V(t)$ as[22,23]

$$V(t) = \frac{k_2}{\log(k_1 t + k_0)} \quad (2)$$

where $k_0$ depends on initial condition as:

$$k_0 = \exp\left(-\frac{k_2}{V_0}\right) \quad$$

**Weight decay model and FN-DAM dynamics**. Many neural network training algorithms are based on solving an optimization problem of the form:[27]

$$\min_{\bar{w}} H(w) = \frac{\alpha}{2} ||\bar{w}|| + \mathcal{L}(\bar{w}) \quad (3)$$

where $\bar{w}$ denotes the network synaptic weights, $\mathcal{L}(\cdot)$ is a loss-function based on the training set and $\alpha$ is a hyper-parameter that controls the effect of the $\mathcal{L}_2$ regularization. Applying gradient descent updates on each element $w_i$ of the weight vector $\bar{w}$ as:

$$w_{i,n+1} - w_{i,n} = -\alpha \eta_n w_{i,n} - \eta_n \frac{\delta \mathcal{L}(\bar{w})}{\delta w_{i,n}} \quad (4)$$

Where the learning rate $\eta_n$ is chosen to vary according to $\eta_n \sim O(1/n)$ to ensure convergence to a local minimum:[30]

The naturally implemented weight decay dynamics in FN-DAM devices can be modeled by applying Kirchhoff's Current Law at the SET and RESET floating gate nodes (see Fig. 1e).

$$C_T \frac{d}{dt}(W_S) + I_{FN}(W_S) = C_C \frac{d}{dt}(V_{SET}) \quad (5)$$

$$C_T \frac{d}{dt}(W_R) + I_{FN}(W_R) = C_C \frac{d}{dt}(V_{RESET}) \quad (6)$$

Where $C_{FG} + C_C = C_T$ is the total capacitance at the floating gate. Taking the difference between the above two equations, we get:

$$C_T \frac{d}{dt}(W_S - W_R) + I_{FN}(W_S) - I_{FN}(W_R) = C_C \frac{d}{dt}(V_{SET} - V_{RESET}) \quad (7)$$

For the differential architecture, $w = W_S - W_R$. Let $V_{train} = V_{SET} - V_{RESET}$, the training voltage calculated by the training algorithm. In addition, $I_{FN}$ is substituted from Eq. 2. Let $C_C/C_T = C_R$, the input coupling ratio:

$$\frac{dw}{dt} = -\frac{(I_{FN}(W_S) - I_{FN}(W_R))}{C_T} + C_R \frac{d}{dt}(V_{train}) \quad (8)$$

$$\frac{dw}{dt} = \frac{-\left(\frac{k_1}{k_2}\right) W_R^2 \exp\left(-\frac{k_2}{W_R}\right) + \left(\frac{k_1}{k_2}\right) W_S^2 \exp\left(-\frac{k_2}{W_S}\right)}{W_R - W_S} w + C_R \frac{d}{dt}(V_{train}) \quad (9)$$

Discretizing the update for a small time-interval $\Delta t$

$$w_{n+1} = w_n + \frac{-\left(\frac{k_1}{k_2}\right) W_R^2 \exp\left(-\frac{k_2}{W_R}\right) + \left(\frac{k_1}{k_2}\right) W_S^2 \exp\left(-\frac{k_2}{W_S}\right)}{W_R - W_S} w_n \Delta t + C_R \Delta V_{train,n} \quad (10)$$

Let $\mu = W_R/W_S$

$$w_{n+1} = w_n - \left(\frac{k_1}{k_2}\right) W_S \exp\left(-\frac{k_2}{W_S}\right) \frac{\mu^2 \exp\left(-\frac{k_2}{W_S}\left(1 - \frac{1}{\mu}\right)\right) - 1}{\mu - 1} w_n \Delta t + C_R \Delta V_{train,n} \quad (11)$$

Assuming that the stored weight (measured in mV) is much smaller than node potential (> 6 V) i.e., $w \ll W_R$ (and $W_R \approx W_S$) and taking the limit ($\mu \to 1$) using L'Hôpital's rule:

$$w_{n+1} = \left(1 - \left(\frac{k_1}{k_2}\right)(2W_S + k_2) \exp\left(-\frac{k_2}{W_S}\right) \Delta t\right) w_n + C_R \Delta V_{train,n} \quad (12)$$

$W_S$ follows the temporal dynamics given in Eq. 1,

$$w_{n+1} = \left(1 - k_1 \left(\frac{2}{\log(k_1 n \Delta t + k_0)} + 1\right)\left(\frac{1}{k_1 n \Delta t + k_0}\right)\right) w_n \Delta t + C_R \Delta V_{train,n} \quad (13)$$

Comparing above equation to Eq. 4, the weight decay factor for FN-DAM system is given as:

$$\alpha \eta_n = k_1 \left(\frac{2}{\log(k_1 n \Delta t + k_0)} + 1\right)\left(\frac{1}{k_1 n \Delta t + k_0}\right) \to O\left(\frac{1}{n}\right) \quad (14)$$

Note that the assumption $w \ll W_{S/R}$ in Eq. (10) makes the mathematical model of the synapse more tractable but is not a requirement for the memory to function. The caveat in relaxing the $w \ll W_{S/R}$ requirement is that the weight decay factor will not scale as $1/n$ during the initial phases of training. However, as shown in Fig. 6d, e for MNIST training, the learning process is able to compensate for this deviation.

**Chip-in-the-loop linear classifier training**. A hybrid hardware-software system was implemented to carry out an online machine learning task. The physical weights ($\bar{w} = [w_1, w_2]$) stored in two FN-DAM devices were measured and used to classify points from a labeled test data set in software. We sought to train a linear decision boundary of the form:

$$f(\bar{x}, \bar{w}) = x_2 + w_1 x_1 + w_0 \quad (15)$$

$\bar{x} = [x_1, x_2]$ are the features of the training set. For each point that was misclassified, the error in the classification was calculated and a gradient of the loss function with respect to the weights was calculated. Based on the gradient information, the weights were updated in hardware by application of SET and RESET pulses via a function generator.

The states of the SET and RESET nodes were measured every 2 s and the weight of each memory cell, $i$, was calculated as:

$$w_i = 1000 \times \left(W_{R,i} - W_{S,i}\right) \quad (16)$$

The factor of 1000 indicates that the weight is stored as the potential difference between the SET and RESET nodes as measured in mV. We followed a stochastic gradient descent method. We defined loss function as:

$$\mathcal{L}_n(\bar{w}) = ReLU\left(1 - y_n f(\bar{x}_n, \bar{w})\right) \quad (17)$$

The gradient of the loss function was calculated as:

$$G_n(\bar{w}) = \frac{\partial \mathcal{L}_n(\bar{w})}{\partial \bar{w}} \quad (18)$$

The weights needed to be updated as

$$w_{n+1} = w_n - \lambda_n G_n(\bar{w}) \quad (19)$$

Here $\lambda_n$ is the learning rate as set by the learning algorithm. The gradient information is used to update FN-DAM by applying control pulses to SET/RESET nodes via a suitable mapping function $T$:

$$V_{train,n} = T(\lambda_n G_n(\bar{w})) \quad (20)$$

Positive weight updates were carried out by application of SET pulses and negative updates via RESET pulses. The magnitude of the update was implemented by modulating the number of input pulses.

**Memory retention model**. In FN-DAM the parameter is stored as difference ($w$) between the dynamical SET ($W_S$) and RESET ($W_R$) nodes as $w = W_s - W_R$. Due to resynchronization between the dynamical nodes, there is a finite time before memory can be read. Moreover, the rate of resynchronization is a function of state of the nodes (Fig. 2 in the main text), therefore, $w$ can have a range of around of 1 V. Assuming 8-bit storage precision to be sufficient for machine learning applications, each memory state is separated from each other by 4 mV. The retention time corresponds to the time it takes for $w$ to reduce from 8 to 4 mV. This time is determined by the time-evolution of the FN-DAM voltage $V(V_0, t)$ which is determined by the parameter array $K = [k_1, k_2]$, and a potential $V_0$ that determines the region of operation (Fig. 2 in the main text). Based on Eq. 2 in the main text $V(V_0, t)$ is given by:

$$V(V_0, t) = \frac{k_2}{\log\left(k_1 t + \exp\left(\frac{k_2}{V_0}\right)\right)} \quad (21)$$

The parameter array $K$ is estimated from experiments which were carried out at room temperature (25 °C). Retention time $T_{ret}$ is calculated by solving the following equation:

$$V(W_R + 0.008, T_{ret}) - V(W_R, T_{ret}) = 0.004 \quad (22)$$

where $W_R$ is varied from 5.5 to 7 V, to simulate different operating regimes. These simulation results are shown in Supplementary Information Fig. 4a. The retention times could then be estimated at different operating temperatures by using the

Arrhenius equation to estimate $k_1$ as a function of temperature as

$$k_1(T) = k_1(T_0)\exp\left(-\frac{E_a}{k}\left(\frac{1}{T} - \frac{1}{T_0}\right)\right) \quad (23)$$

For instance, in our retention estimation shown in Supplementary Information Fig. 4a, we assumed activation energy $E_a = 0.6eV$, $k$ is the Boltzmann's constant (8.617 eV/K) and $T_0 = 25\,°C$. Also, Supplementary Information Fig. 4b. shows that the retention model given by Eqs. 19 and 20 matches the measured results at 100 °C.

A larger FN-DAM array chipset was used for chip-in-the-loop neural network training on the Fisher Iris dataset. The chipset contains 128 individually selectable and programmable tunneling devices. By utilizing them in pairs, 64 synaptic weights could be implemented. Of the 64 FN-DAM elements, 51 elements were used to store the training weights of a three-layer neural network model (5 units corresponding to 4 input features and one for the bias term in the input layer, 7 units (including 1 bias unit) in the hidden layer and 3 units in the output layer). A five-fold cross-validation analysis was performed by splitting the Iris dataset into sets of 30 points each. Over 5 training sessions, each of the 5 sets was used to validate the model trained on the remaining 4 sets (120 points). The training was conducted over 20 epochs. In each epoch, the training set was randomly shuffled, and a batch size of 10 points was selected. During each batch update, weight updates were calculated through backpropagation using stochastic gradient descent. These weight updates were carried out in hardware through the application of SET/RESET pulses to the corresponding memory cell. Retention of the trained MLP parameters was verified using bake experiments as described in Supplementary Information Section VIII and summarized in SI Figs. 9, 10.

**FN-DAM based CNN Implementation**. The performance of FN-DAM model was compared to that of a standard network model. A 15-layer convolutional neural network was trained on the MNIST dataset using the MATLAB Deep Learning Toolbox. For each learnable parameter in the CNN, a software FN-DAM instance corresponding to that parameter was created. In each iteration, the loss of the network function and gradients were calculated. The gradients were used to update the weights via Stochastic Gradient Descent with Momentum (SGDM) algorithm. The updated weights were mapped onto the FN-DAM array. The weights in the FN-DAM array were decayed according to Eq. 14. These weights were then mapped back into the CNN. This learning process was carried on for 9 epochs. In the 10th epoch, no gradient updates were performed. However, the weights were allowed to decay for the last epoch (note that in the standard CNN case, the memory is static). A special case with a 0.1% randomly assigned mismatch in the floating gate parameters ($k_1$ and $k_2$) was also implemented.

## Data availability
All the software and experimental data used for generating the figures have been deposited in a public repository (https://doi.org/10.6084/m9.figshare.19295474)[46].

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

## Acknowledgements

This work was supported in part by National Science Foundation (ECCS: 1935073), Office of Naval Research (N00014-16-1-2426, N00014-19-1-2049), and by National Institutes of Health (1R21EY028362-01).

## Author contributions

S.C. and D.M. came up with the concept of FN-DAM. D.M. and S.C. designed the hardware and simulation experiments, D.M. designed the 24 element FN-DAM chipset; M.R. designed the 64 element FN-DAM chipset; D.M. and M.R. conducted the simulation and hardware experiments; K.A. designed and ran the CNN experiments. S.C. provided supervision on all tasks. All authors contributed towards writing and proofreading the manuscript.

## Competing interests

The authors declare no competing interests.
