## [Peer Review File · Nature Communications]

Reviewers' Comments:

Reviewer #1:

Remarks to the Author:

The paper is well-written. The natural dynamics of synapses is shown to regularize dynamics of weights during training approximately. Low voltage ($\sim 3V$) based disturbance of the dynamics produces weights approximately at the millivolt level. These two effects produce low energy training. As a proof, a 2-synapse training is demonstrated experimentally. Later, the models are extracted and a large network is demonstrated in simulations for MNIST.

Though the concept is interesting and well presented, the claims of the paper are questionable both theoretically as well as from the perspective of demonstration.

Firstly, the retention dynamics i.e. tunneling current (I) is fundamentally internal electric field (E) driven i.e. I/E^2 is proportional to $\log(E)$. This is a straight line i.e. it changes smoothly. In comparison the work exploits a sharp transition in tunneling current which is shown in Fig. 1 as a schematic. Plotting out the analytical equation exactly will reveal that such a step function (shown in Fig. 1) does not exist. Hence, designing around the transition edge - which is the concept of the paper appears to be difficult to accept. On hindsight, traditional Flash memory avoids this by operating far away from this transition i.e. stable V_t - so that retention is 10 years. However, the idea of operating near the transition to get first time evolving followed by long retention places the device where neither time evolution can be controlled nor retention is sufficient.

Second, 1mV difference in threshold voltage is used to measure the state. This level of sensitive measurement is possible for individual devices. However, 1mV difference (even below transistor threshold where current is exponentially dependent on V_t) produces 4% current difference even in the ideal case of 60 mV/decade of subthreshold slope (SS) of a MOSFET (which is ideal). Realistic 120mV/decade SS would produce a 2% current change. Above the transistor threshold, this will produce a 0.1% current difference. In a memory array of 1000 devices (e.g. in MNIST first layer), such measurement depends upon a very tight distribution of memory states. Any disturbance during reading (read-disturb) will be very problematic. So an access transistor will be necessary to isolate bitcells and measure each bitcell accurately. Now, typical logic access transistors have standard deviation in V_t of approximately 20mV in a technology. So how does one read 1mV shift in bitcells where access transistors (which are more controlled than memory) are blurring the distribution with 20x more V_t variation. Thus, array operation must be proposed and verified - albeit in circuit simulations with variability.

Thirdly, possibly due to this difficulty, only a two synapse training is demonstrated in hardware for only 500s (when the V_t is still time evolving as per authors themselves). Given that the authors show a chip photo, there must be multiple production quality devices (unlike say emerging devices built in labs). Typically, 8-bit precision or equivalent gradual analog weight change over a range is required to train an MNIST network (<https://www.nature.com/articles/s41467-018-04933-y>). This level of control must be demonstrated through range and precision for digital or SNR/variability for analog weights. Hence, at least more statistics equivalent to a simple Fisher Iris (16x4 weight matrix) is needed for hardware demonstration - i.e. ~ 64 different synapses and multiple cycles. These weights must be trained equivalent to the 2-synapse example shown by the authors. Relative and absolute weight dynamics must be shown for (1) short term dynamics followed by (2) long term retention in V_t vs log time graph. The experimental weights compared against ideal/expected weights in training simulations. The inference and training phases should be shown as a clear demarcation experimentally. Retention should be supported further with high temperature testing (e.g. 250 degree C for 6 hours). This will at least demonstrate the synapse performance without the complications of creating an array. Here, it is important to also comment on the size and scalability of the device. The authors' devices are fabricated in large 0.5um technology. Larger devices are less variable. But noise will increase with scaling. The reviewer is tempted to request a comment on how 1mV signal impacts scalability to advanced nodes.

Authors have shown temperature dependence on the w . Authors are requested to define what is w e.g. is it weight change per pulse? The drift in w is shown to be approximately $\pm 5mV$, however, w itself is of range of $\sim mVs$. Will the temperature variation disturb the weight state of the device?

This is not even a 10 year retention loss emulation.

Based on these comments, the work seems challenging from both theoretically and practically (from a systems concept and demonstration). They require further demonstration as pointed out above. Below are some further questions.

Further questions :

Characterization of synapse:

SNR - The modelling requires the $w \ll W_S/R$, but is this also a requirement for the volatile/non-volatile action of the synapse to be present physically on the devices? If so, how do the authors plan to make a scalable compute in-memory network with sub 10 mV signal synapses in the presence of noise and variability on-chip?

How would these small signals be measured in an array (e.g. In STT RAM, even 2X change in state is only good enough for digital operation), then how such a small SNR is sufficient? On chip sense and measurement circuits have a 20 mV margin requirement to distinguish signals. How will the sub mV signals respond to disturbs in the circuit?

Initialization - How do we achieve small difference synchronization, can some data be shown on initialisation and the level of synchronisation achieved.

FN requires large voltages to begin with. Yet the programming that affects the dynamics uses a small voltage $\sim 3V$ for a longer time. How do we know that this is charge storage in floating gate and not in traps?

Is retention timescale V_t dependent?

Reviewer #2:

Remarks to the Author:

Here are my brief comments about the paper, mainly about the claim on reducing training energy. I can appreciate the idea from this manuscript on the different strategies to trade off data retention for programming energy in the training and inference phases.

However, this method requires that the training outcome is unknown to claim any benefit so its application is limited on analog training engines. It would not grant energy savings if the training is done offline and the programming targets of the devices are known. In this case, one should always use the programming method that provides sufficient data retention.

Although the idea seems to be promising, I don't find the manuscript showing the benefit with clear experimental data. In page 5 line 191 it claimed Fig. 5 g-h showed significant reduction in energy dissipation during training, but the graph doesn't really show that and the authors should plot the data clearly and provide quantitative results on this "reduction in energy dissipation". In Fig.5g, the standard net seems to spend lower energy as iteration increases (compared with FN-DAM with mismatch). It is also not clear what the "standard" net is for the comparison. Although in page 11 line 342, there is the paragraph about the "FN-DAM based CNN Implementation", the hardware implementation of the "standard" net that was used for the comparison in Fig. 5g, h is still not clear to me.

My biggest concern though on this paper is related to reduction to practice. It would be good if the authors could demonstrate their claims in a simple array.

Reviewer #3:

Remarks to the Author:

In this manuscript, the authors present the application of a Fowler-Nordheim tunneling-based dynamic analog memory (FN-DAM) as synaptic device for deep neural network tasks. The study evidences that this device exhibits dynamic memory states enabling a weight evolution in time which can be exploited to significantly decrease the energy consumption during the neural network training procedure.

In my opinion, this work addresses an interesting research topic. However, I think that the manuscript currently evidences weak points in terms of technical results, clarity, and readability. The list of my comments/questions/suggestions is reported below.

1) Starting from device and architecture concepts already proposed in a work recently published in Nature Communications (D. Mehta et al, DOI: 10.1038/s41467-020-19292-w), the authors address a new application of the FN-DAM device as synaptic element for deep neural networks. However, I think that the experimental characterization and application sections require more detailed explanations and results. Also, the discussion section should be improved by a more quantitative comparison, mainly in terms of energy consumption, with some of emerging memristive technologies mentioned in the work.

2) I think that these points of Introduction section are not clear:

- Rows 24-26: this sentence should be better explained with more details/references.
- What is the size of synaptic DAM array used in this work? More details on the structure of the array shown in Fig. 1(d) should be provided in the manuscript.

3) I think that the authors should improve the clarity and add more details in the sections focused on device concept and operation as follows:

- Row 122: the desynchronization degree w is defined as WS-WR, but it is obtained as WR-WS in Figs. 3(a-d). I also noted that w is proportional to WR-WS in eq. 14. Please, could the authors clarify this point?
- Fig 2 caption: more details on operation conditions and meaning of color dots should be also specified in the figure and corresponding caption.

4) I think that the authors should add more details in the experimental sections as follows:

- information on the experimental setup used in this work should be added.
- more details on the applied voltage pulse schemes used for experiments in Fig. 4 should be better specified:

Fig. 4a: what are the number and width of applied voltage pulses?

Fig. 4b: what are the amplitude and width of applied voltage pulses?

Fig. 4c: what are the amplitude and width of applied voltage pulses?

Fig. 4d: Why did the authors limit the temperature study to 40°C?

5) Regarding the section on neural network applications, the following points should be addressed:

- The CNN architecture should be better described (only 15 layers information is not clear). How many convolutional and fully-connected layers does the neural network architecture include? What is the number of updated weights (size of programmed DAM-based array)?
- The submission of MNIST dataset for CNN classification task should also be mentioned in the text. Did the authors use all the 60000/10000 grayscale 28x28 images for network training/test?
- An additional figure showing the final weights programmed in FN-DAM-based array would be useful for improving the description of network implementation.
- A more extensive study of the impact of floating-gate parameters mismatch ($> 0.1\%$) on network accuracy could provide more details on FN-DAM-based network reliability.
- An additional figure showing the confusion matrix with accuracy values for each digit class could improve the CNN implementation description.
- Quantitative comparisons in terms of energy/area/speed with similar network architectures based on memristive synaptic devices such as RRAM or PCM should be discussed to better evidence the potential of this device concept for high-performance hardware DNN tasks.

6) I think that the manuscript readability should be improved. Many figure references in the text are wrong and the figure captions should include more details.

- Row 91/95: Fig. 1(g) instead of Fig. 1(f)
- Row 83/110: Fig. 1(d) instead of Fig. 1(e).
- Fig. 1(e) should be described in the manuscript.
- Row 160: Fig. 4(c) instead of Fig. 4(a).

7) A check of typos should also be carried out to further improve the manuscript readability

- Row 28: 'based on'

- Row 29: 'Spin-Transfer Torque'
- Row 51: 'be adaptive'
- Row 110: 'programmed'
- Row 132: 'show'
-

Response to the reviewers

We thank the reviewers for insightful comments and positive feedback. We have addressed the concerns raised by the reviewers and summarized them here (also highlighted in the main manuscript). In addition to including new experimental results in the main and supplementary sections of the manuscript, we have also corrected typographical and notational errors in the revised manuscript.

Reviewer #1 (Remarks to the Author):

The paper is well-written. The natural dynamics of synapses is shown to regularization dynamics of weights during training approximately. Low voltage ($\sim 3V$) based disturbance of the dynamics produces weights approximately at the millivolt level. These two effects produce low energy training. As a proof, a 2-synapse training is demonstrated experimentally. Later, the models are extracted and a large network is demonstrated in simulations for MNIST.

Though the concept is interesting and well presented, the claims of the paper are questionable both theoretically as well as from the perspective of demonstration.

- 1. Firstly, the retention dynamics i.e. tunneling current (I) is fundamentally internal electric field (E) driven i.e. I/E^2 is proportional to $\log(E)$. This is a straight line i.e. it changes smoothly. In comparison the work exploits a sharp transition in tunneling current which is shown in Fig. 1 as a schematic. Plotting out the analytical equation exactly will reveal that such a step function (shown in Fig. 1) does not exist. Hence, designing around the transition edge - which is the concept of the paper appears to be difficult to accept. On hindsight, traditional Flash memory avoids this by operating far away from this transition i.e. stable V_t - so that retention is 10 years. However, the idea of operating near the transition to get first time evolving followed by long retention places the device where neither time evolution can be controlled nor retention is sufficient.*

Response 1: The reviewer is correct in noting that the tunneling current is continuous with no sharp changes. However, the sharp transition shown in Fig. 1 is the change in the potential difference across the tunneling barrier. Since Fig. 1(g) is just an illustration of the concept, we have not included the transients during the period when the control (SET or RESET) pulse is applied. However, in our previous work [24] we have described in detail how the FN tunneling current can be modulated by any arbitrary input signal. Based on those prior results, in Fig. R1 we show how a SET pulse can create different transients. As shown in Fig. R1, the application of a SET pulse pushes the voltage of the tunneling node (W_S) to a level where electron tunneling-rate is significantly higher than the RESET node (denoted by its voltage W_R). This causes the SET node to desynchronize with respect to the RESET node and the degree of desynchronization acts as a memory. Fig. R1 also shows that after the cessation of the SET pulse, the W_R and W_S resynchronize with each other as depicted by $w = W_S - W_R$. The degree of desynchronization and the rate of resynchronization are both a function of state of the system (voltage previously stored on the memory), which we are exploiting for this work as a Fowler-Nordheim dynamic analog memory (FN-DAM). We have clarified this in introduction section of the revised manuscript.

Figure R1: Output measured from the FN device when subjected to an input pulse. During the positive half of the input pulse, the tunneling rate increases and desynchronizes the SET device with respect to the RESET device. Reproduced from [24].

Figure R2: (a) Training using conventional non-volatile memory with fixed energy-barrier height separating the sequential memory states W_{n-1} and W_n ; and (b) Training using FN-DAM where the energy-barrier height is adapted to transition from low memory retention during learning to retention times similar to conventional floating-gate memories during inference.

We agree with the reviewer that traditional FLASH memory achieves high memory retention time by using a stable V_t or a high energy barrier separating different memory states. The same concept is true for other non-volatile memories as well, as is illustrated in Fig. R2(a) where each sequential memory states W_{n-1} and W_n are separated by an energy barrier of constant height ΔE . This attribute, however, does not exploit the fact that learning/training is a dynamical process and therefore a traditional approach has to dissipate ΔE Joules for every parameter update. The key idea/concept of this paper is to make the height of the energy barrier separating the sequential memory states adaptive, as shown in Fig. R2(b). During the initial phases of training the height of the energy barrier ΔE_1 is small and the transition between memory states is driven by the dynamics of learning or minimizing the energy-loss function $L(\mathbf{W})$. The challenge in implementing the adaptive barrier height is to ensure that the time-constants for memory leakage is large enough to support meaningful learning. This aspect has been successfully demonstrated in this paper using FN-DAM.

However, note that as the training evolves (or parameters are learned), the height of the energy barrier ΔE increases and asymptotically FN-DAM transitions to a conventional FLASH memory. In this regard the retention at the end of training would be similar to that of a FLASH memory. We have clarified this in the introduction section and in the discussion section of the revised manuscript.

2. *Second, 1mV difference in threshold voltage is used to measure the state. This level of sensitive measurement is possible for individual devices. However, 1mV difference (even below transistor threshold where current is exponentially dependent on V_t) produces 4% current difference even in the ideal case of 60 mV/decade of subthreshold slope (SS) of a MOSFET (which is ideal). Realistic 120mV/decade SS would produce a 2% current change. Above the transistor threshold, this will produce a 0.1% current difference. In a memory array of 1000 devices (e.g. in MNIST first layer), such measurement depends upon a very tight distribution of memory states. Any disturbance during reading (read-disturb) will be very problematic. So an access transistor will be necessary to isolate bitcells and measure each bitcell accurately. Now, typical logic access transistors have standard deviation in V_t of approximately 20mV in a technology. So how does one read 1mV shift in bitcells where access transistors (which are more controlled than memory) are blurring the distribution with 20x more V_t variation. Thus, array operation must be proposed and verified - albeit in circuit simulations with variability.*

Response 2: We agree that if the proposed FN-DAM is used as a static memory, measuring 1mV differences to distinguish between different memory states would be challenging, especially if device mismatch is taken into account. However, note that value stored on the FN-DAM array is updated inside a learning loop that minimizes a system level objective function $L(W)$ in Fig. R2(a). Therefore, the effect of any static mismatch across memory cells will get calibrated out during the process of training, as has been shown by us and other research groups [38,39]. What is important for this calibration process to be successful is that the memory update be monotonic with respect to error-gradient and the precision of the update be greater than 12 bits [26,36]. Both of these requirements are met by FN-DAM due to the physics of electron tunneling. In fact, the effect of calibration due to learning can be seen in the FN-DAM neural network training (Fig. 6 d,e) where the final accuracy is independent of the initial choice of the FN-DAM state and the mismatch in FN-DAM device characteristics.

The effect of blurring, due to the presence of thermal noise is in fact beneficial for training the neural network since it helps in overcoming artifacts due to local minima. Once the FN-DAM has transitioned to a non-volatile state (during inference), the effect of blurring reduces because of the presence of energy barrier which is significantly higher than thermal fluctuations. The effect of blurring due to measurement noise can be reduced by averaging and longer measurement time.

Read-disturbances during training could also be potentially calibrated for if the FN-DAM memory is updated inside the learning loop. However, in this case, the calibration procedure will be more involved because of the non-linear coupling between the readout circuit and the memory elements storing the weight parameters. To reduce the effect of read disturbance, in our implementation we have capacitively decoupled the readout circuit from the memory as shown in Fig S1 in the Supplementary Material. We have also conducted read-disturbance experiments (as suggested by the reviewer), where twelve FN-DAM memory elements were randomly accessed every minute for 1000 cycles and the

relative change in weight was measured after every read. The measured result shown in Fig. R3 verifies that read-disturb in our implementation of FN-DAM is random and the magnitude is less than the precision of the update and measurement. We have included Fig. R3 and brief description of the experiment in the Supplementary Material.

Figure R3: Read disturbance reflected as change in weight parameters measured from 12 FN-DAM devices over 1000 read cycles. Each color in the figure represents one FN-DAM device.

3. *Thirdly, possibly due to this difficulty, only a two-synapse training is demonstrated in hardware for only 500s (when the V_t is still time evolving as per authors themselves). Given that the authors show a chip photo, there must be multiple production quality devices (unlike say emerging devices built in labs). Typically, 8-bit precision or equivalent gradual analog weight change over a range is required to train an MNIST network (<https://www.nature.com/articles/s41467-018-04933-y>). This level of control must be demonstrated through range and precision for digital or SNR/variability for analog weights. Hence, at least more statistics equivalent to a simple Fisher Iris (16x4 weight matrix) is needed for hardware demonstration - i.e. ~ 64 different synapses and multiple cycles. These weights must be trained equivalent to the 2-synapse example shown by the authors. Relative and absolute weight dynamics must be shown for (1) short term dynamics followed by (2) long term retention in V_t vs log time graph. The experimental weights compared against ideal/expected weights in training simulations. The inference and training phases should be shown as a clear demarcation experimentally. Retention should be supported further with high temperature testing (e.g. 250 degree C for 6 hours). This will at least demonstrate the synapse performance without the complications of creating an array. Here, it is important to also comment on the size and scalability of the device. The authors' devices are fabricated in large 0.5 μm technology. Larger devices are less variable. But noise will increase with scaling. The reviewer is tempted to request a comment on how 1mV signal impacts scalability to advanced nodes.*

Response 3:

- As per the suggestion of the reviewer, we have also designed an experiment where the FN-DAM array was operated at 100 °C. Note that compared to standard reliability testing of non-volatile memories where the chipsets are baked offline and then the retention measurements could be performed under standard operating temperature, for testing FN-DAM, the data needs to be measured continuously under high temperature condition. This is because FN-DAM is a dynamical memory which stores information in the degree of temporal desynchronization between two dynamical systems. Therefore, performing continuous measurements under 250°C operating condition would have required us to significantly change our test setup (read-out circuitry would be required to be functional and temperature compensated) and cabling requirements. However, the required reliability information can also be inferred from continuous measurements at 100°C. In FN-DAM the parameter is stored as difference (w) between the dynamical SET (W_S) and RESET (W_R) nodes as $w = W_S - W_R$. Due to resynchronization between the dynamical nodes, there is a finite time before which memory can be read. Moreover, the rate of resynchronization is a function of state of the nodes (Fig. 2 in the main text), therefore, w can have a range of around of 1V. Assuming 8-bit storage precision to be sufficient for machine learning applications, each memory state is separated from each other by 4 mV. Therefore, the retention time corresponds to the time it takes for w to reduce from 8 mV to 4 mV. This time is determined by the time-evolution of the FN-DAM voltage $V(V_0, t)$ which is determined by the parameter array $K = [k_1, k_2]$, and a potential V_0 that determines the region of operation (Fig. 2 in the main text). Based on equation (2) in the main text $V(V_0, t)$ is given by:

$$V(V_0, t) = \frac{k_2}{\log\left(k_1 t + \exp\left(\frac{k_2}{V_0}\right)\right)}$$

The parameter array K is estimated from experiments which were carried out room temperature (25 °C). Retention time T_{ret} is calculated by solving the following equation:

$$V(W_R + 0.008, T_{ret}) - V(W_R, T_{ret}) = 0.004$$

where W_R is varied from 5.5 to 7 V, to simulate different operating regimes. These simulation results are shown in Fig. R4a.

Next, we estimated the effect of temperature on the retention times. We used the Arrhenius equation to estimate k_1 as a function of temperature:

$$k_1(T) = k_1(T_0) \exp\left(-\frac{E_a}{k} \left(\frac{1}{T} - \frac{1}{T_0}\right)\right)$$

We assumed activation energy $E_a = 0.6$ eV, k is the Boltzmann's constant (8.617eV/K) and $T_0 = 25^\circ C$.

Fig. R4a shows retention times for 60°C and 100°C which we verified using experiments conducted at 100°C. Fig. 4b shows the measured results where the weights stored in 12 FN-DAM devices kept at 100°C were measured over a duration of 15 hours. The baseline drift due to the memory read-out circuits were first calibrated during the first 400 min and used to zero out the dynamical response of each of the FN-DAM device. Then, at 400 min time instant a SET pulse (3.3V for 1 second duration) was applied to all the memory devices which programmed all the device to a specific memory state.

The degree of desynchronization was continuously measured and is plotted in Fig. 4b. The resynchronization process is accurately predicted by the model at 100°C (Fig. R4b inset)

Figure R4: a) Simulated retention time as a function of SET/RESET node voltage for different operating temperatures. b) Measurement and modeling results from 12 FN-DAM devices desynchronized at 100°C .

- The key requirement for implementing FN-DAM in any process is the availability of thick-oxide (oxide thickness greater than 10nm). This feature is available in most advanced fabrication nodes, especially since the Input/Output transistors require a thicker oxide layer. Also, note that FN-DAM is compatible with the FLASH process, and hence these memory devices can also be implemented on an advanced NOR FLASH architectures.
- Following the reviewer’s suggestion, we have implemented a realistic training protocol on a larger FN-DAM chipset. This chipset contains 128 individually selectable and programmable tunneling devices. By utilizing them in pairs, 64 synaptic weights could be implemented. Of the 64 weights, we used 49 weights to train a network model (with 7 units in the hidden layer) on the Fisher Iris dataset. For each run, the IRIS dataset with 150 points was randomly split into 110 points for training, 20 points for validation and 20 points were kept aside for testing. The training was conducted over 2 epochs. In each epoch, the training set was randomly shuffled, and a batch size of 10 points was selected. During each batch update, weight updates were calculated through backpropagation with stochastic gradient descent. These weight updates were carried out in hardware through application of SET/RESET pulses to the corresponding memory cell. The measured results are shown in Fig. R5 which we have also included in Fig. 6 of the revised manuscript.

Figure R5: Experimental training on Fisher Iris dataset: a) Trained network on Fisher Iris dataset. Thickness of connections between units indicate magnitudes of learned weights. Blue (red) connection indicates positive (negative) weight. b) 5-fold cross-validation accuracy of model over 20 epochs for training set (120 points) and validation set (30 points) c) Total pulses required in implementing weight update for entire synaptic array during each epoch. d) Energy per unit capacitance expended in updating the weights. e) Confusion matrix for the trained network.

4. *Authors have shown temperature dependence on the w . Authors are requested to define what is w e.g. is it weight change per pulse? The drift in w is shown to be approximately $\pm 5mV$, however, w itself is of range of $\sim mVs$. Will the temperature variation disturb the weight state of the device? This is not even a 10 year retention loss emulation.*

Response 4: Since we have run a new set of experiment at 100C, we can better characterize the retention time as shown in Fig. 4(d). Note that FN-DAM is a dynamic memory, and we are exploiting its desynchronization or leakage properties to improve the energy-efficiency for machine learning training. However, when the training converges or the parameters stored on FN-DAM transition to a non-volatile state, we project that we can achieve XX years of retention in this state.

Based on these comments, the work seems challenging from both theoretically and practically (from a systems concept and demonstration). They require further demonstration as pointed out above. Below are some further questions. Further questions :

Characterization of synapse:

5. *SNR - The modelling requires the $w \ll W_{S/R}$, but is this also a requirement for the volatile/non-volatile action of the synapse to be present physically on the devices? If so, how do the authors plan to make a scalable compute in-memory network with sub 10 mV signal synapses in the presence of noise and variability on-chip?*

Response 5: No, the requirement $w \ll W_{S/R}$ make the mathematical modeling of the synapse tractable but is not a requirement for the memory to function. However, note that the typical values of $W_{S/R}$ in FN-DAM

is about 6V, and w is in the range of 1V. The updates to w are in the order of microvolts, so the condition $w < W_{SR}$ is satisfied. The caveat in relaxing the $w \ll W_{SR}$ requirement is that the weight decay factor (α_n) will not scale as $1/n$ during the initial phases of training, but as we show in Fig. 6(d)-(e) for MNIST training, the learning process is able to compensate for this deviation. We have clarified this in the Methods section of the revised manuscript where we derive the $1/n$ scaling.

6. *How would these small signals be measured in an array (e.g. In STT RAM, even 2X change in state is only good enough for digital operation), then how such a small SNR is sufficient? On chip sense and measurement circuits have a 20 mV margin requirement to distinguish signals. How will the sub mV signals respond to disturbs in the circuit?*

Response 6: As Fig. R3 shows, readout disturb noises are less than +/- 200 μ V and thus readout of mV signals is feasible. However, the minimum signal that can be measured by the on-chip sense and measurement circuits are governed by the readout speeds and allowed power dissipation limits. In this regard, we agree with the reviewer that design of high-speed read-out circuits to differentiate sub-mV changes will be challenging, given energy-efficiency constraints. SI Fig. 4 shows the tradeoffs between readout SNR, bandwidth and power dissipated.

7. *Initialization - How do we achieve small difference synchronization, can some data be shown on initialisation and the level of synchronisation achieved.*

Response 7: The synchronization of FN based dynamical systems has been shown across a range of previous works [23], across devices located on the same die and across devices located on different dies. In [24 – Fig. 2], we reported an initialization procedure that can reliably synchronize the two FN dynamical system within +/- 200 μ V. Note this accuracy was limited by the precision of the measurement/read-out circuitry.

8. *FN requires large voltages to begin with. Yet the programming that affects the dynamics uses a small voltage ~ 3V for a longer time. How do we know that this is charge storage in floating gate and not in traps?*

Response 8: The reviewer is correct to note that FN requires large voltages. For the 13 nm oxide thickness implemented here, the potential difference across the oxide needs to be over 8V. During initialization, we set the FG voltage to around 8V relative to the substrate by setting the bulk at 21V. The bulk voltage is then removed and electrons start flowing ‘slowly’ into the floating gate due to the electric field created across the oxide. Due to the 3V input signal, the FG voltage increases to around 10V (after attenuation by the capacitive divider). It is in this phase that the FN current of SET node significantly outpaces the RESET node. As the FG node discharges, a larger and larger programming voltage is required to reach the target threshold of 10V. But as the baseline tunneling rates of the two nodes are lower, it takes longer for the RESET node to resynchronize, leading to large retention times. In principle, at high-voltages the dynamics could be affected by three types of tunneling: FN tunneling, direct tunneling and trap-assisted tunneling (TAT). Since the tunneling barrier thickness is greater than 12nm, the contribution of direct tunneling is negligible. Trap assisted tunneling on the other hand depends on the quality of the oxide and the Si-SiO₂ interface. Since the measured currents and dynamics show an excellent agreement with $1/\log(t)$ characteristics that can only arise due to FN tunneling, we can assume that the TAT current is also negligible.

9. Is retention timescale V_t dependent?

Response 9: Both V_t and retention time scale depend on device parameters like oxide thickness and thus are indirectly related to each other.

Reviewer #2 (Remarks to the Author):

Here are my brief comments about the paper, mainly about the claim on reducing training energy. I can appreciate the idea from this manuscript on the different strategies to trade off data retention for programming energy in the training and inference phases. However, this method requires that the training outcome is unknown to claim any benefit, so its application is limited on analog training engines. It would not grant energy savings if the training were done offline and the programming targets of the devices are known. In this case, one should always use the programming method that provides sufficient data retention.

Response: We agree with the reviewer that the key benefit of using FN-DAM is to reduce energy dissipation during training, especially in analog and compute-in-memory architectures. This is what has been highlighted in Fig. 1 in the main manuscript. If FN-DAM is used only as a non-volatile analog storage of trained parameter, then the configuration is identical to the floating-gate memories which has already been shown for energy-efficient inference. We have clarified this in the introduction and the discussion section of the revised manuscript. We have also included an additional inset in Fig. 1 to clarify Fig R2(b) – please see response 1 to reviewer 1.

For other non-volatile memories as well, as is illustrated in Fig. R2(a) each sequential memory states \mathbf{W}_{n-1} and \mathbf{W}_n are separated by an energy barrier of constant height ΔE . This attribute, however, does not exploit the fact that learning/training is a dynamical process and therefore a traditional approach has to dissipate ΔE Joules for every parameter update. The key idea/concept of this paper is to make the height of the energy barrier separating the sequential memory states adaptive, as shown in Fig. R2(b). During the initial phases of training the height of the energy barrier ΔE_1 is small and the transition between memory states is driven by the dynamics of learning or minimizing the energy-loss function $L(\mathbf{W})$. The challenge in implementing the adaptive barrier height is to ensure that the time-constants for memory leakage is large enough to support meaningful learning. This aspect has been successfully demonstrated in this paper using FN-DAM. However, note that as the training evolves (or parameters are learned), the height of the energy barrier ΔE increases and asymptotically FN-DAM transitions to a conventional FLASH memory.

Although the idea seems to be promising, I don't find the manuscript showing the benefit with clear experimental data. In page 5 line 191 it claimed Fig. 5 g-h showed significant reduction in energy dissipation during training, but the graph doesn't really show that and the authors should plot the data clearly and provide quantitative results on this "reduction in energy dissipation". In Fig.5g, the standard net seems to spend lower energy as iteration increases (compared with FN-DAM with mismatch). It is also not clear what the "standard" net is for the comparison. Although in page 11 line 342, there is the paragraph about the "FN-DAM based CNN Implementation", the hardware implementation of the "standard" net that was used for the comparison in Fig. 5g, h is still not clear to me. My biggest concern though on this paper is related to reduction to practice. It would be good if the authors could demonstrate their claims in a simple array.

Response: We have demonstrated the energy-efficiency benefits for a 64 size FN-DAM array and for a classification task using the Fisher Iris dataset. Please see the response to reviewer 1 as well. The new measurement result in Fig. 6(a)-(c) show that the energy consumption throughout training can be made relatively constant, as has been claimed in this work. We have clarified and updated Fig. 5g,h (now Fig. 6d,c in the revised manuscript) by Fig. R6 below. For comparison we have used energy per bit metric for RRAM (100 fJ/bit – Ref. 17 in main text), which is the lowest reported energy consumption among the emerging non-volatile memories. The figure shows that FN-DAM based neural network dissipates significantly less energy during the initial part of the training compared to the RRAM based neural network. However, for FN-DAM neural network the energy dissipated per bit increases as training progresses; but the number of parameter updates also decreases. As a result, the energy consumed remains approximately constant with time, which was the goal of this paper (please see Fig. 1(c)).

Figure R6: Modeling experiments comparing the energy efficiency of different CNN variants trained using the MNIST dataset: (a) loss-function corresponding to different training iterations and different CNN implementations; and (b) total energy dissipated to update the training parameter for different compute-in-memory technologies.

Reviewer #3 (Remarks to the Author):

In this manuscript, the authors present the application of a Fowler-Nordheim tunneling-based dynamic analog memory (FN-DAM) as synaptic device for deep neural network tasks. The study evidences that this device exhibits dynamic memory states enabling a weight evolution in time which can be exploited to significantly decrease the energy consumption during the neural network training procedure.

In my opinion, this work addresses an interesting research topic. However, I think that the manuscript currently evidences weak points in terms of technical results, clarity, and readability. The list of my comments/questions/suggestions is reported below.

1. *Starting from device and architecture concepts already proposed in a work recently published in Nature Communications (D. Mehta et al, DOI: 10.1038/s41467-020-19292-w), the authors address a new application of the FN-DAM device as synaptic element for deep neural networks. However, I think that the experimental characterization and application sections require more detailed explanations and*

results. Also, the discussion section should be improved by a more quantitative comparison, mainly in terms of energy consumption, with some of emerging memristive technologies mentioned in the work.

Response: We thank the reviewer for this comment. We have included more detailed explanation of the experiments in the Methods section and in the Supplementary materials. We have also updated Fig. 6 (d-e) in the revised manuscript which shows comparison with RRAM based neural network implementation which has the lowest reported energy consumption (100 fJ/bit – Ref. 17), among the emerging non-volatile memories.

2. *I think that these points of Introduction section are not clear:*

i Rows 24-26: this sentence should be better explained with more details/references.

ii What is the size of synaptic DAM array used in this work? More details on the structure of the array shown in Fig. 1(d) should be provided in the manuscript.

Response: We have included more details and references for Rows 24-26. We have used two FN-DAM arrays for our experiments. The simple perceptron experiments used an array comprising of two banks of 12 FN-DAM devices. Each of the devices in the bank received the same input signals. Therefore, we could only train a classifier with only two weights. However, this array is useful to characterize the reliability of the system with respect to device programming and read-out. The new set of experiments on the Fisher Iris classification task used an array of 64 FN-DAM devices. This chipset contains 128 individually selectable and programmable tunneling devices (as shown in Fig. 1 in the supplementary information). By utilizing these devices in pairs, 64 synaptic weights could be implemented. Of the 64 weights, we used 49 weights to train a network model (with 7 units in the hidden layer) on the Fisher Iris dataset.

3. *I think that the authors should improve the clarity and add more details in the sections focused on device concept and operation as follows:*

i Row 122: the desynchronization degree w is defined as $W_S - W_R$, but it is obtained as $W_R - W_S$ in Figs. 3(a-d). I also noted that w is proportional to $W_R - W_S$ in eq. 14. Please, could the authors clarify this point?

Response: We chose $w = W_R - W_S$ so that a SET pulse leads to a positive change in the stored value w . This has been corrected in the manuscript. We have also included additional details on the device concept and operation in the supplementary information section of the manuscript. Note that some of the details of floating-gate layout and fabrication have been previously reported in [23] so in the supplementary information we succinctly present the device concept for the sake of completeness.

ii Fig 2 caption: more details on operation conditions and meaning of color dots should be also specified in the figure and corresponding caption.

Response: We included more details in Fig. 2 caption.

4. *I think that the authors should add more details in the experimental sections as follows:*

i Information on the experimental setup used in this work should be added.

ii *More details on the applied voltage pulse schemes used for experiments in Fig. 4 should be better specified:*

iii *Fig. 4a: what are the number and width of applied voltage pulses?*

iv *Fig. 4b: what are the amplitude and width of applied voltage pulses?*

v *Fig. 4c: what are the amplitude and width of applied voltage pulses?*

Response: These details have been added to the figure captions. We have included more information about the experimental setup in the methods section and section 1 in the supplementary material.

vi *Fig. 4d: Why did the authors limit the temperature study to 40°C?*

Response: Please see response to reviewer 1 – Response 3, where we have included measurement results corresponding to 100°C operation which also has been used for retention estimation.

5. *Regarding the section on neural network applications, the following points should be addressed:*

i *The CNN architecture should be better described (only 15 layers information is not clear). How many convolutional and fully-connected layers does the neural network architecture include? What is the number of updated weights (size of programmed DAM-based array)?*

Response: We have included the following details about the CNN in the supplementary section of the revised manuscript. The CNN consists of the following layers.

Layer	Name	Description	Activations	Learnables
1	Image Input	28×28×1 images	28*28*1	0
2	Convolution	20 5×5×1 convolutions with stride 1	24*24*20	520
3	Batch Normalization	Batch normalization with 20 channels	24*24*20	40
4	ReLU	ReLU	24*24*20	0
5	Convolution	20 3×3×20 convolutions with stride 1	24*24*20	3620
6	Batch Normalization	Batch normalization with 20 channels	24*24*20	40
7	ReLU	ReLU	24*24*20	0
8	Max	Pooling	12*12*20	0
9	Convolution	40 3×3×20 convolutions with stride 1	12*12*40	7240
10	preluLayer	Parametric ReLU with 40 channels	12*12*40	40
11	Convolution	20 3×3×40 convolutions with stride 1	12*12*20	7220
12	Batch Normalization	Batch normalization with 20 channels	12*12*20	40
13	ReLU	ReLU	12*12*20	0
14	Fully Connected	10 fully connected layer	10	28810
15	Softmax	softmax	10	0

This table has been added to the SI. The network was constructed in MATLAB using Deep Learning toolbox and was trained using Stochastic Gradient Descent with Momentum. For simulating FN-DAM array, only the weights in the Fully Connected layer were trained.

- ii *The submission of MNIST dataset for CNN classification task should also be mentioned in the text. Did the authors use all the 60000/10000 grayscale 28x28 images for network training/test?*

Response: All 60000/10000 images were used for training/testing the networks.

- iii *An additional figure showing the final weights programmed in FN-DAM-based array would be useful for improving the description of network implementation.*

Response: In the supplementary material section, we have included the network topology (Fig. R5) for Fisher Iris task. Thickness of connections between units indicate magnitudes of learned weights. Blue (red) connection indicates positive (negative) weights. The final values of the programmed weights for the MNIST task is available through a data repository.

- iv *A more extensive study of the impact of floating-gate parameters mismatch ($> 0.1\%$) on network accuracy could provide more details on FN-DAM-based network reliability.*

Response: Details of how the parameter mismatch was introduced has been added to the Methods section of the manuscript and in the supplementary section. However, note that value stored on the FN-DAM array is updated inside a learning loop that minimizes a system level objective function $L(W)$ in Fig. R2(a). Therefore, the effect of any static mismatch across memory cells will get calibrated out during the process of training, as has been shown by us and other research groups [40,41]. What is important for this calibration process to be successful is that the memory update be monotonic with respect to error-gradient and the precision of the update be greater than 12 bits [26]. Both of these requirements are met by FN-DAM due to the physics of electron tunneling. In fact, the effect of calibration due to learning can be seen in the FN-DAM neural network training (Fig. 6 d,e) where the final accuracy is independent of the initial choice of the FN-DAM state and the mismatch in FN-DAM device characteristics. We have included a discussion related to this in the revised manuscript.

- v *An additional figure showing the confusion matrix with accuracy values for each digit class could improve the CNN implementation description.*

Response: These have been added to the supplementary material section of the manuscript - also shown here.

Validation Error 0.9%

0	973		1	1			1	1	3		99.3%	0.7%
1		1126	1	2				5	1		99.2%	0.8%
2	1	1	1028					1	1		99.6%	0.4%
3				1009					1		99.9%	0.1%
4			1		974				2	5	99.2%	0.8%
5				7		883	1			1	99.0%	1.0%
6	5	1	1	1	1	3	942		4		98.3%	1.7%
7		1	5	2	1			1017		2	98.9%	1.1%
8	1		1	1		1			969	1	99.5%	0.5%
9	1		2		5	2		2	5	992	98.3%	1.7%

99.2%	99.7%	98.8%	98.6%	99.3%	99.3%	99.8%	99.1%	98.3%	99.1%
0.8%	0.3%	1.2%	1.4%	0.7%	0.7%	0.2%	0.9%	1.7%	0.9%
0	1	2	3	4	5	6	7	8	9

Predicted Class

vi *Quantitative comparisons in terms of energy/area/speed with similar network architectures based on memristive synaptic devices such as RRAM or PCM should be discussed to better evidence the potential of this device concept for high-performance hardware DNN tasks.*

Response: We have included a few statement of comparison in the discussion section of the revised manuscript. Note that for the comparison to be fair, we have used the same learning algorithm for all the synaptic devices. We have also updated Fig. 6 (d-e) in the revised manuscript which shows comparison with RRAM based neural network implementation. RRAM has one of the lowest reported energy consumption (100 fJ/bit – Ref. 17), among the emerging non-volatile memories.

6. *I think that the manuscript readability should be improved. Many figure references in the text are wrong and the figure captions should include more details.*

- i - Row 91/95: Fig. 1(g) instead of Fig. 1(f)
- ii - Row 83/110: Fig. 1(d) instead of Fig. 1(e).
- iii - Fig. 1(e) should be described in the manuscript.
- iv - Row 160: Fig. 4(c) instead of Fig. 4(a).

Response: These have been corrected.

7. *A check of typos should also be carried out to further improve the manuscript readability*

- i - Row 28: 'based on'
- ii - Row 29: 'Spin-Transfer Torque'

iii - Row 51: *'be adaptive'*

iv - Row 110: *'programmed'*

v - Row 132: *'show'*

Response: These have been corrected.

Reviewers' Comments:

Reviewer #1:

Remarks to the Author:

The authors have made some improvements in terms of demonstrating experimental training on a larger problem and including read noise data. However, a few responses were not aligned properly to the reviewer questions and some big concerns are not addressed. We highlight these:

Major Concerns:

1. The authors have demonstrated read disturbance in change of weight to be approx. 200uV. The read disturb data is shown for individual memory elements. However, in a network level application, there will be inherent device to device variability (or mismatch) in the peripheral circuitry including read sense amplifiers. An accurate measurement-read of all the synaptic weight will be dependent upon the sensitivity of the access transistors and variability of read circuitries across the network. How do authors plan to differentiate signals due to peripheral read circuitry offsets since the signal strength is too small (~ 10 mV)? What should be the area penalty for such accurate circuits compared to standard techniques?
2. The synapses are used in training as well as inference phase, with device's non-volatility and retention in the inference phase. However, we are concerned how they will operate at this transition edge of retention (volatile to non-volatile) and get good inference retention. For this we suggest, In the Fisher Iris demonstration, a clear demarcation should be shown between the training and inference phase. Then, after training is complete then all the devices in the network should be stressed at high temperatures ~ 250 degrees Celsius for a few hours (standard retention measurement) and then reading out the state at the room temperature to see if they retained the trained weight. This request was made earlier but the authors misunderstood this as read to be performed at 250 degree C. However, we just need a 250 degree 6 hour bake with pre and post bake read at room temperature.
3. The author's are requested to comment on the scalability of the devices for a network application. The devices presented in the paper are fabricated in large 0.5 um node technology. For larger network applications, the devices will have to be scaled down. The variability amongst scaled devices will be significant as opposed to large area devices. This will affect (1) the signal levels of the proposed device, and (2) the read capability of scaled devices. A quantitative analysis with simulation based study should be done with advanced node devices incorporating practical noise and variability effects. It may be insufficient to qualitatively say that noise and variability are not a problem for training applications which have a feedback loop and need to escape local minima. Quantitative simulation to show the impact of noise and variability on training performance is necessary.

Some Other Concerns:

1. The transition asked in question 1 was not about abrupt change in state on application of Write pulse. The transition being referred to is the transition from volatile to non-volatile nature of the synapses. This transition cannot be abrupt. Operation at the edge of low retention is problematic for long-term inference. Hence, in question 3, it was asked to produce the short term and long-term dynamics of the weights. This data was not produced.
2. W.r.t question 9, the question asks whether the retention timescales of the memory are a function of its state, V_t . In the same device do we get different retention depending on what is the V_t value currently? Higher V_t vs Lower V_t - does the state decay with the same timescale.

Reviewer #2:

None

Reviewer #3:

Remarks to the Author:

The authors addressed the questions and comments reported by this Reviewer by improving the technical quality of the manuscript. However, in my opinion minor revisions are still necessary to further improve the paper clarity and readability, which is affected by many typos.

1) Lines 123-143 main text: Fig. 2a with each inset (a1, a2, a3) should be briefly mentioned with reference to 3 different initialization conditions. Also, as already evidenced in the previous review (question #3i), w is always defined as WS-WR in the manuscript (e.g., line 129,149,174, etc.) even if it is obtained as WR-WS in Fig. 3c (I'd expect that $\Delta w/WS$ increases after SET input pulse application) and Eq. 14. I suggest clarifying this point by correcting either the text or Fig. 3c to avoid misunderstanding. In addition, I suggest adding the percentage changes reported in the main text in Fig. 2b,2c,2d using an x-axis step of 10 s.

List of typos:

- Main text

Line 29: 'These include two-terminal memristive devices such as ...' instead of 'These include the crossbar memristor based on';

Line 30: 'Spin-Transfer Torque Magnetic RAM (STT-MRAM)' instead of 'Spin-Torque Transfer RAM';

Line 31: 'Ferroelectric field-effect transistor (FeFET)' instead of 'ferroelectric field-effect transistor-based RAM (FeRAM)'. Unlike FeFET, which is a three-terminal memory device, ferroelectric RAM (FeRAM) should be included in the class of two-terminal memory devices.

Line 35: 'in RRAM devices' instead of 'memristive devices';

Line 86: 'for implementing the circuit of Fig. 1e';

Line 124: Reference to Fig. 1c seems wrong. 'as described in Fig. 1g' should replace 'as described in Fig. 1c';

Line 150: 'a sequence of 100 ms-long 3V SET and RESET pulses was applied' could replace 'a sequence of 100 ms SET and RESET pulses were applied';

Line 169: a reference to 10 ms pulse-length may be mentioned to provide more detail;

Line 185: 'which is expected to vary' in place of 'which a expects varies';

Line 209: 'when the MLP neural network is trained using the Fisher Iris dataset' instead of 'when the neural network trained using the Fisher Iris dataset';

Line 232: ML acronym is already defined in the introduction of manuscript;

Line 411-412: 'which were carried out at room temperature (25°C). Retention time T_{ret} ';

Line 475-567: Many upper cases in the titles of references are missing. I suggest checking it in all the reference list;

Line 596: Fig. 4 – (d) should be added in the corresponding figure;

Line 617: captions (d) and (e) of Fig. 6 are inverted. I also suggest distinguishing the initialization of Figure 5 (Synaptic memory for perceptron neural network task) from Figure 6 (Synaptic memory for deep neural network tasks).

- Supplementary Information:

Line 17-21: some typos should be fixed. REF should be specified;

Lines 43/49: SI Fig. 3a/ SI Fig. 3b;

Line 59: SI Fig. 4a shows the retention times for different T (25°C, 60°C, 100°C);

Line 75: (a) and (b) are missing in SI Fig. 4;

Lines 107-110: SI Fig. 7a and 7b should be mentioned and separately discussed in Section VII.

RESPONSE TO REVIEWER COMMENTS

We thank the reviewers for their constructive comments which we have addressed in the revised manuscript. Below we provide a point-by-point response to each of the reviewer's comments. All changes in the main manuscript and in the supplementary information have been highlighted in blue.

Reviewer #1 (Remarks to the Author):

Comment 1. *The authors have demonstrated read disturbance in change of weight to be approx. 200 μ V. The read disturb data is shown for individual memory elements. However, in a network level application, there will be inherent device to device variability (or mismatch) in the peripheral circuitry including read sense amplifiers. An accurate measurement-read of all the synaptic weight will be dependent upon the sensitivity of the access transistors and variability of read circuitries across the network. How do authors plan to differentiate signals due to peripheral read circuitry offsets since the signal strength is too small (~ 10 mV)? What should be the area penalty for such accurate circuits compared to standard techniques?*

Response:

With respect to the complexity and area-penalty of the read-out circuitry, in this work we have used a voltage buffer (source follower) to read the state of the FN-DAM cell. However, a current mode readout could also be used to differentiate 10mV changes. In particular, if the read-out transistor is biased in weak-inversion, then the exponential dependence between the gate voltage and the drain current could be used to amplify the change in voltage. We have previously used this method [REF1] to access the state of floating-gate memories and the method also allowed for temperature compensation and calibration. In [REF2] we also reported an active feedback-based method to improve the resolution of the voltage-mode read-out for a high-density floating-gate memory array. Note that in both these designs, there will be a trade-off with the resolution of the read-out and the read-out speed. We have included a brief description regarding this in the discussion section of the revised manuscript.

[REF1] M. Gu, S. Chakrabarty, "A Varactor-driven, Temperature Compensated CMOS Floating-gate Current Memory with 130ppm/K Temperature Sensitivity", IEEE Journal of Solid-State Circuits, vol. 47, no: 11, pp. 2846-2856, Nov. 2012.

[REF2] L. Zhou, S. Chakrabarty, "A 7-Transistor-Per-Cell, High-Density Analog Storage Array with 500 μ V Update Accuracy and Greater Than 60dB Linearity", IEEE Symposium on Circuits and Systems (ISCAS 2014), Melbourne, Australia, 2014.

We agree with the reviewer that the peripheral read-out circuitry (voltage or current-mode) will introduce gain and offset errors when reading the FN-DAM voltages. However, we envision that during training and inference each of memory cell will be queried by the same read circuitry. Since the training is performed using a chip-in-the-loop procedure, the variability across the memory cell and readout circuitry will be compensated for. As an example, in Fig. 6(d) and (e), we show a proof of concept where we could train a neural network that incorporates 10% mismatch across the FN-DAM cells. The result shows minimal degradation in accuracy. Also, note that even though the single update could be in mV, the final trained weights could be in the range of 100mVs. This is highlighted in the results presented in response to Comment 2 (below). We acknowledge that a significant mismatch between the SET and RESET nodes of the same cell could have a negative impact on performance of the network, especially if the network is not properly initialized. However, chip-in-the-loop training should be able to compensate for these artifacts as well.

Comment 2. *The synapses are used in training as well as inference phase, with device’s non-volatility and retention in the inference phase. However, we are concerned how they will operate at this transition edge of retention (volatile to non-volatile) and get good inference retention. For this we suggest, In the Fisher Iris demonstration, a clear demarcation should be shown between the training and inference phase. Then, after training is complete then all the devices in the network should be stressed at high temperatures ~ 250 degrees Celsius for a few hours (standard retention measurement) and then reading out the state at the room temperature to see if they retained the trained weight. This request was made earlier but the authors misunderstood this as read to be performed at 250 degree C. However, we just need a 250 degree 6 hour bake with pre and post bake read at room temperature.*

Response:

We agree with the reviewer that the data retention times (or the cells’ volatility) is a smooth function with respect to the dynamic state of the memory (SET and RESET voltages). This is also shown in SI Fig. 4a. We initially operate the FN-DAM array in the volatile region and trade off data retention for lower energy consumption. Once the network has been trained, it is not necessary to wait for the system to reach the non-volatile regime. Because of the differential architecture of the FN-DAM, both the SET and RESET nodes discharge/decay into the slow-tunneling regime and the discharge-rate is approximately constant across all the memory cells. In this case, the performance of the learning algorithm (or a neural network) that normalizes the weights will remain robust with minimal degradation in recognition accuracy. We have verified this by conducting the baking experiment as suggested by the reviewer.

We trained a neural network, shown in SI Fig. 7(a) on the Fisher’s Iris dataset using FN-DAM as storage for network parameters. Post-training, we transferred our chip into a baking oven which was set to 225°C. Note that this is the maximum temperature setting for the baking oven (Quincy lab Model 40) set at 225 °C. Care was taken in transporting the chipsets to prevent any electrostatic discharge (ESD) issues. After 6 hours of baking, the chips were taken out, weights were read-out and the classification accuracy of the neural network was evaluated. Fig. 1 a, below compares the weights stored on the FN-DAM cells, before and after baking. The result in Fig. 1(a) shows that even though all the post-bake weights exhibit a decay with respect their pre-bake values, when normalized ($w_{norm} = w/\|w\|_1$), both the pre-bake and post-bake values remain relatively invariant.

Fig. 1 (a) Weights stored on FN-DAM memory cells after training a neural network on the Fisher-Iris dataset, before baking (Pre) and after baking (Post); and (b) normalized weights before baking and after baking.

Fig. 2 Training and test accuracy obtained using the pre-bake and post-bake values of weights stored on the FN-DAM.

Fig. 2. compares the training and test accuracy obtained using weights stored on the FN-DAM before baking and after baking. The result shows that while the training accuracy reduces nominally (97.5% to 95%), the test accuracy remains unchanged. Note that after the bake, the respective SET and RESET voltages (W_S and W_R) decays such that the FN-DAM enters the high-retention regime. We have included these new results and discussions in the Supplementary Information Section VIII.

Comment 3. *The authors are requested to comment on the scalability of the devices for a network application. The devices presented in the paper are fabricated in large 0.5 μm node technology. For larger network applications, the devices will have to be scaled down. The variability amongst scaled devices will be significant as opposed to large area devices. This will affect (1) the signal levels of the proposed device, and (2) the read capability of scaled devices. A quantitative analysis with simulation based study should be done with advanced node devices incorporating practical noise and variability effects. It may be insufficient to qualitatively say that noise and variability are not a problem for training applications which have a feedback loop and need to escape local minima. Quantitative simulation to show the impact of noise and variability on training performance is necessary.*

Response: We have included comments regarding the scalability of the devices in the Discussions section of the paper. We agree with the reviewer that for a large network, the device form-factor need to be reduced which affect the variability. However, in our prior work [23,34] we have shown that the dynamics of the FN-DAM cell (in steady-state) is determined primarily by the gate-oxide thickness (t_{ox}) and a weak function of the form-factor. We include Fig. 3. here (from [34]) to highlight this.

Fig. 3. Results reported in [23,34] which shows that the dynamics of the core FN-DAM cell is robust to device form-factor (and hence mismatch).

Also, in our previous work we have verified the functionality basic FN-DAM cell across different CMOS processes (130nm and 180nm). We include Fig. 4. here which shows that the dynamical systems model equation (19) is valid, except for when the oxide-thickness is less than 7nm when direct tunneling current also contributes to the gate-leakage. However, most sub-10nm process have an option of using thicker gate-oxides ($t_{\text{ox}} > 10\text{nm}$), in which case FN-DAM could be easily fabricated in those processes.

Fig. 4. Measured results showing that the FN-DAM dynamics is scalable across processes with gate-oxide-thickness $t_{\text{ox}} > 10\text{nm}$.

To understand the effect of FN-DAM device variability on the accuracy of ML architecture, in Table 1, we have trained an FN-DAM based convolutional neural network assuming 0.1% mismatch in FN-DAM model parameters for the MNIST task. The results in Table 1 show a minimal degradation in test accuracy.

Some Other Concerns:

Comment (a). *The transition asked in question 1 was not about abrupt change in state on application of Write pulse. The transition being referred to is the transition from volatile to non-volatile nature of the synapses. This transition cannot be abrupt. Operation at the edge of low retention is problematic for long-term inference. Hence, in question 3, it was asked to produce the short term and long-term dynamics of the weights. This data was not produced.*

Response: We agree with the comment that the transition from volatile to non-volatile nature of the synapse cannot be abrupt. And in Fig. 2(b)-(d) we show the short-term and long-term dynamics of the memory where 2(b) corresponds to a regime with lower retention and 2(c) correspond to the regime with higher retention.

We would also like to point out that the retention property of FN-DAM can also be switched from low to high using the voltage V_{prog} shown in schematic in Fig. 5, which is common to all FN-DAM cells. After the training is completed, V_{prog} could be lowered which will push all the FN-DAM cells into the high-retention regime. Note that in this case, the mismatch in the coupling capacitor C_{prog} will determine the inference accuracy. In the high-retention regime, the leakage is expected to be similar to EEPROM since we are using a floating-gate technology as well.

Fig 5: Circuit implementation of the FN-DAM cell with read-out and programming circuitry. The global voltage V_{prog} is highlighted and can be used to shift the state W of all the FN-DAM cells.

Comment (b). *W.r.t question 9, the question asks whether the retention timescales of the memory are a function of its state, V_t . In the same device do we get different retention depending on what is the V_t value currently? Higher V_t vs Lower V_t - does the state decay with the same timescale.*

Response: There is a weak-dependency between the rate of decay and V_t . However, as shown in the bake experiment in Fig. 1-2, the weak-dependency can be compensated by the training algorithm.

Reviewer #3 (Remarks to the Author):

The authors addressed the questions and comments reported by this Reviewer by improving the technical quality of the manuscript. However, in my opinion minor revisions are still necessary to further improve the paper clarity and readability, which is affected by many typos.

Comment 1. Lines 123-143 main text: Fig. 2a with each inset (a1, a2, a3) should be briefly mentioned with reference to 3 different initialization conditions. Also, as already evidenced in the previous review (question #3i), w is always defined as WS-WR in the manuscript (e.g., line 129,149,174, etc.) even if it is obtained as WR-WS in Fig. 3c (I'd expect that $\Delta W/WS$ increases after SET input pulse application) and Eq. 14. I suggest clarifying this point by correcting either the text or Fig. 3c to avoid misunderstanding. In addition, I suggest adding the percentage changes reported in the main text in Fig. 2b,2c,2d using an x-axis step of 10 s.

Response: We have made the suggested changes. For. Fig. 2b-d, adding steps of 10s looked a bit cluttered, so we added only a step of 25s.

Comment 2. *List of typos:*

- *Main text*

Line 29: 'These include two-terminal memristive devices such as ...' instead of 'These include the crossbar memristor based on';

Line 30: 'Spin-Transfer Torque Magnetic RAM (STT-MRAM)' instead of 'Spin-Torque Transfer RAM';

Line 31: 'Ferroelectric field-effect transistor (FeFET)' instead of 'ferroelectric field-effect transistor-based RAM (FeRAM)'. Unlike FeFET, which is a three-terminal memory device, ferroelectric RAM (FeRAM) should be included in the class of two-terminal memory devices.

Line 35: 'in RRAM devices' instead of 'memristive devices';

Line 86: 'for implementing the circuit of Fig. 1e';

Line 124: Reference to Fig. 1c seems wrong. 'as described in Fig. 1g' should replace 'as described in Fig. 1c';

Line 150: 'a sequence of 100 ms-long 3V SET and RESET pulses was applied' could replace 'a sequence of 100 ms SET and RESET pulses were applied';

Line 169: a reference to 10 ms pulse-length may be mentioned to provide more detail;

Line 185: 'which is expected to vary' in place of 'which a expects varies';

Line 209: 'when the MLP neural network is trained using the Fisher Iris dataset' instead of 'when the neural network trained using the Fisher Iris dataset;

Line 232: ML acronym is already defined in the introduction of manuscript;

Line 411-412: 'which were carried out at room temperature (25°C). Retention time Tret';

Line 475-567: Many upper cases in the titles of references are missing. I suggest checking it in all the reference list;

Line 596: Fig. 4 – (d) should be added in the corresponding figure;

Line 617: captions (d) and (e) of Fig. 6 are inverted. I also suggest distinguishing the initialization of Figure 5 (Synaptic memory for perceptron neural network task) from Figure 6 (Synaptic memory for deep neural network tasks).

- Supplementary Information:

Line 17-21: some typos should be fixed. REF should be specified;

Lines 43/49: SI Fig. 3a/ SI Fig. 3b;

Line 59: SI Fig. 4a shows the retention times for different T (25°C, 60°C, 100°C);

Line 75: (a) and (b) are missing in SI Fig. 4;

Lines 107-110: SI Fig. 7a and 7b should be mentioned and separately discussed in Section VII.

Response: We thank the reviewer for pointing out the typos which we have corrected in the revised manuscript and supplementary information.

Reviewers' Comments:

Reviewer #3:

Remarks to the Author:

The authors addressed all the questions and comments by improving the previous version of the manuscript. No additional changes are required from my side.

We thank the reviewers for their insightful comments and we gladly note that they found our responses satisfactory.

Reviewer Comments

Reviewer #3 (Remarks to the Author):

The authors addressed all the questions and comments by improving the previous version of the manuscript. No additional changes are required from my side.